# Transposable elements mediate genetic effects altering the expression of nearby genes in colorectal cancer

Nikolaos M. R. Lykoskoufis [1,2,3,4] ✉, Evarist Planet[5], Halit Ongen [1,2,3,6], Didier Trono [5,6] & Emmanouil T. Dermitzakis [1,2,3,6] ✉

Transposable elements (TEs) are prevalent repeats in the human genome, play a significant role in the regulome, and their disruption can contribute to tumorigenesis. However, TE influence on gene expression in cancer remains unclear. Here, we analyze 275 normal colon and 276 colorectal cancer samples from the SYSCOL cohort, discovering 10,231 and 5,199 TE-expression quantitative trait loci (eQTLs) in normal and tumor tissues, respectively, of which 376 are colorectal cancer specific eQTLs, likely due to methylation changes. Tumor-specific TE-eQTLs show greater enrichment of transcription factors, compared to shared TE-eQTLs suggesting specific regulation of their expression in tumor. Bayesian networks reveal 1,766 TEs as mediators of genetic effects, altering the expression of 1,558 genes, including 55 known cancer driver genes and show that tumor-specific TE-eQTLs trigger the driver capability of TEs. These insights expand our knowledge of cancer drivers, deepening our understanding of tumorigenesis and presenting potential avenues for therapeutic interventions.

Understanding the mechanisms underlying tumorigenesis has been one of the main research questions in cancer biology. While somatic mutations, chromosomal rearrangements and gene amplification are the three main hallmarks driving cancer progression, they are unable to provide a complete explanation of tumorigenesis. Recent discoveries have demonstrated that transposable elements (TEs) have contributed to the evolution of gene regulation and can alter the landscape of gene expression in development and disease[1–5]. Transposable elements (TEs) are interspersed repeats that contribute more than half of the human genome. TEs, more specifically TE-embedded regulatory sequences (TEeRS) are broadly active during the phases of genome reprogramming that occur in the germline and the early embryo, and then controlled by epigenetic mechanisms that still allow their finely orchestrated participation in biological events as diverse as brain development, immune responses, and metabolic control. The aberrant re-activation of TEeRSs is observed under certain conditions and disease states, notably cancer[6–8]. Transcription is defined by the coordinated activity of regulatory elements which are modulated by genetic variation. Thus, we speculate that transposable element expression is influenced by regulatory non-coding variants, also called expression Quantitative Trait Loci (eQTLs), known to contribute to the onset and progression of complex diseases like cancer[9,10]. To build on this concept, we set out to analyze the interplay between regulatory variants (eQTLs), transposable elements and gene expression to characterize the genetic perturbation of TE and gene expression in cancer. In this paper, we integrated genome-wide genotyping data (genotype array) and transcriptomic profiles (bulk RNA-sequencing) from the Systems Biology of Colorectal Cancer (SYSCOL) cohort

[1]Department of Genetic Medicine and Development, University of Geneva Medical School, 1211 Geneva, Switzerland. [2]Institute for Genetics and Genomics in Geneva (iGE3), University of Geneva, 1211 Geneva, Switzerland. [3]Swiss Institute of Bioinformatics, 1211 Geneva, Switzerland. [4]NGS-AI JSR Life Sciences, Route de la Corniche 3, 1066 Epalinges, Switzerland. [5]School of Life Sciences, Ecole Polytechnique Fédérale de Lausanne (EPFL), 1015 Lausanne, Switzerland. [6]These authors contributed equally: Halit Ongen, Didier Trono, Emmanouil T. Dermitzakis. ✉e-mail: nikolaos.lykoskoufis@gmail.com; emmanouil.dermitzakis@unige.ch

comprising of 275 and 276 normal and tumor samples, respectively. We discovered thousands of eQTLs regulating TE expression both in normal and tumor as well as many tumor specific TE-eQTLs likely driven by methylation changes. Notably, we observed that tumor-specific TE-eQTLs show a greater enrichment of transcription factors compared to shared TE-eQTLs suggesting that TEs more specifically regulated in tumor. Furthermore, by using Bayesian networks, we discovered thousands of TEs acting as mediators of genetic effects, significantly altering the expression of nearby genes, including many known cancer driver genes and showed that tumor-specific TE-eQTLs trigger the driver capability of TEs. Overall, we show that TEs are important mediators of genetic effects onto nearby genes, specifically in cancer, highlighting, their importance during tumorigenesis.

## Results

### Quantifying transposable elements (TEs) and gene expression

To measure the expression of TEs in CRC, we examined transcriptomes obtained by RNA-seq from 275 normal and 276 CRC samples from the SYSCOL cohort[11]. We quantified TE and gene expression using an in-house curated TE annotation list originating from the RepBase database[12] that contains approximately 4.6 million individual TE loci. These annotations were merged with gene annotation from ensembl (v19). Filtering for uniquely mapped reads (Methods) to obtain robust estimates of TE expression resulted in 50,921 TEs and 17,430 genes

(protein coding and lincRNAs). We observed that the majority of expressed TEs present in our dataset are SINEs (Alu and MIR), LINEs (L1 and L2) as well as different subfamilies of Long Terminal Repeats (LTRs) and DNA transposons. However, when we looked at the proportion of expressed TEs per subfamily, SVA and ERVK were most prominent (Fig. 1a, Supplementary Fig. 1). Additionally, we used available data from Encode[13] and miRbase[14] to generate a list of regulatory regions and discovered that 13,590 expressed TEs overlapped with at least one previously identified regulatory element. We also discovered that expressed TEs are significantly enriched for most regulatory regions, except for enhancers, compared to non-expressed TEs (Supplementary Data 1; Fig. 1b, Supplementary Fig. 2) highlighting their potential role in gene expression regulation.

### Transposable elements are under strong genetic control

Using TE expression quantifications and genotype data we first sought to assess the impact of inter-individual genetic variation on TE expression. We conducted cis-eQTL analysis followed by a forward backward stepwise conditional analysis (Methods) and discovered a total of 10,231 and 5199 TE-eQTLs as well as 6955 and 1552 gene eQTLs in normal and tumoral tissue, respectively (Supplementary Figs. 3 and 4; Supplementary Data 2, 3). Similarly to gene-eQTLs, TE-eQTLs displayed stronger effects and density closer to the transcription start site (TSS) in both normal and tumor samples (spearman rho = −0.33,

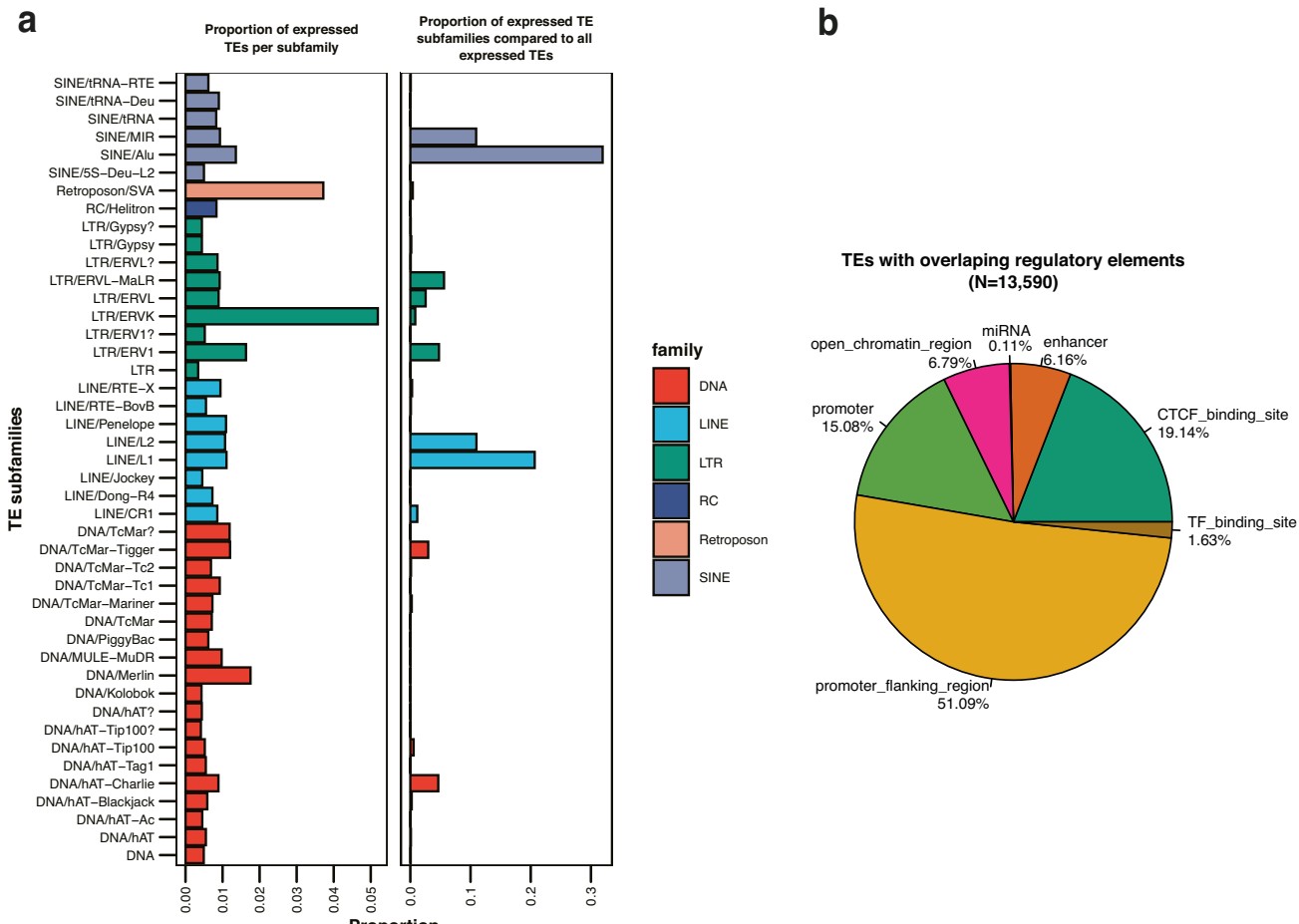

**Fig. 1 | Description of quantified TEs. a** Barplot showing the proportion of uniquely mapped and quantified TE subfamilies in our dataset. **b** Pie chart showing the proportion of TEs with different types of regulatory elements within their sequence. We uniquely mapped and quantified 50,921 TEs. The majority of them are SINEs from the Alu and MIR family, L1 and L2 TEs from the LINE family and

different subfamilies of LTRs as well as some DNA transposons. When we looked at the proportion of expressed TE per subfamily, we observed that SVA and ERVK are most prominent. Additionally, 13,590 out of the 50,921 TEs contain regulatory elements within sequence.

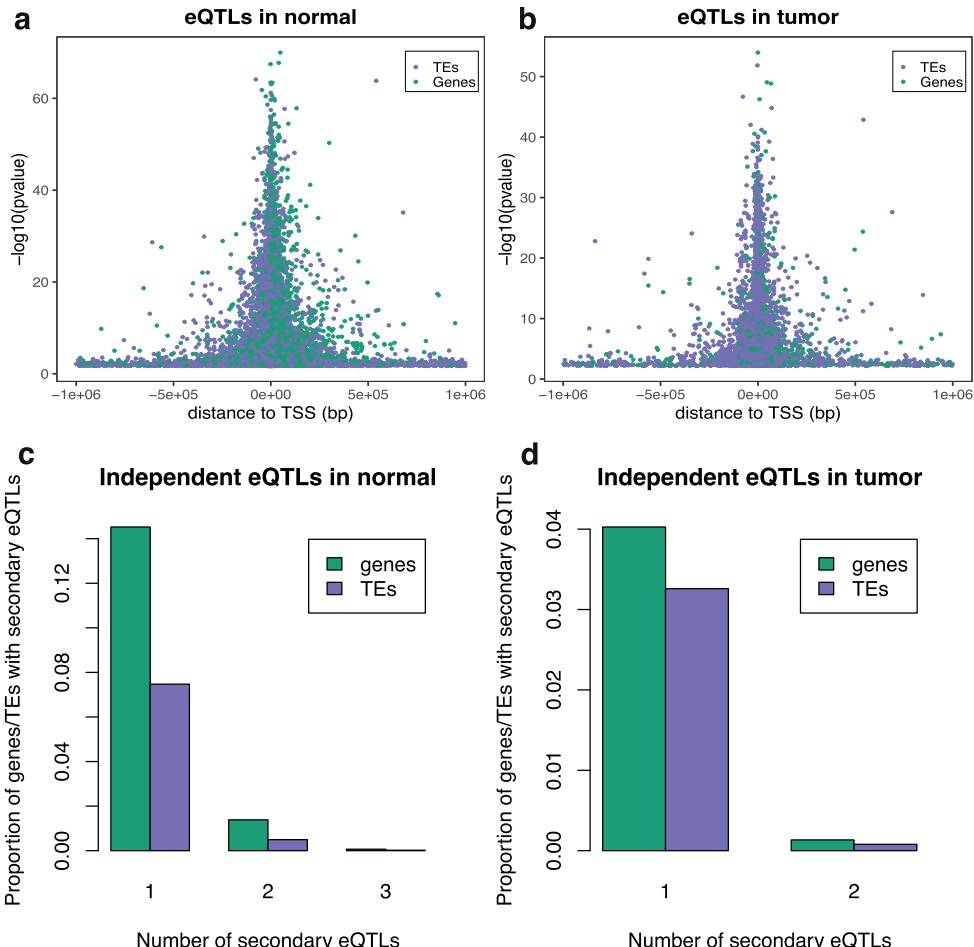

**Fig. 2 | cis- eQTL discovery.** eQTL variant distance to TSS in **a** normal and **b** in tumor. We observe stronger eQTL effect close to the transcription start site of TE and genes in both normal (two-sided Wilcoxon, $p = 2.4e{-}12$) and tumor (two-sided Wilcoxon, $p = 4.8e{-}7$). Number of secondary eQTLs for TEs and genes in **c** normal and **d** tumor. Gene eQTLs have more functionally independent eQTLs per gene than TEs do. Source data are provided as a Source Data file.

$p < 2.2e{-}16$ in normal, spearman rho $= -0.25$, $p < 2.2e{-}16$ in tumor) (Fig. 2a, b), yet were more proximal to the TSS compared to gene-eQTLs (two-sided Wilcoxon $p = 2.4e{-}12$ in normal; two-sided Wilcoxon $P = 4.8e{-}07$ in tumor; Supplementary Fig. 5). We observed that TEs displayed fewer independent eQTLs per TE than genes (Fig. 2c, d) while the minor allele frequencies of TE- and gene-eQTL variants were similar (Supplementary Fig. 6). Proximal distance of TE-eQTLs to TSS and the smaller number of independent signals per TE could be due to smaller evolutionary time of TE regulatory landscapes in the human genome compared to genes, making proximal effects much more likely.

To corroborate our findings, we used external datasets to replicate our eQTL discoveries. We downloaded available data from GTEx for colon transverse ($N = 174$) and TCGA for colon adenocarcinoma (TCGA-COAD, $N = 251$). We processed both datasets in a similar way as we did with the SYSCOL dataset (methods 2-3). Not all variant-feature pairs were present in the GTEx colon transverse dataset after all filtering steps. Out of the 10,231 TE-eQTLs and 6955 gene-eQTLs discovered in normal, 8380 (82%) and 5930 (85%) TE- and gene-eQTLs, respectively were present in the dataset and could be replicated. From the 5199 TE- and 1552 gene-eQTLs discovered in SYSCOL tumor, only 3221 (62%) TE- and 1164 (75%) gene-eQTLs were present in the TCGA-COAD dataset. We observe a high replication of our original results (Supplementary Fig. 7A–C; Supplementary Data 4) in normal (pi1 TE-eQTLs $= 0.831$ pi1 gene-eQTLs $= 0.686$) and tumor (pi1 TE-eQTLs $= 0.884$; pi1 gene-eQTLs $= 0.783$) (Supplementary Fig. 7D–F; Supplementary Data 5) corroborating our findings.

Given previously established roles of tumor-specific gene-eQTLs in tumorigenesis[11], we aimed next at investigating whether tumor-specific TE-eQTLs could similarly contribute as cancer driving factors. To this end, we used linear mix models with an interaction term between variant and tissue (normal/tumor). We discovered that 429 (8%) of the tumor TE-eQTLs are tumor-specific and 1697 (24%) of the normal TE-eQTLs are normal-specific, with 525 TE-eQTLs shared between both settings (Fig. 3a; Supplementary Data 6). For genes, we found 117 (%) tumor gene-eQTLs to be tumor-specific and 902 (%) normal gene-eQTLs to be normal-specific, of which 175 were shared (Supplementary figure 8A; Supplementary Data 7). Shared TE- and gene-eQTLs were closer to the TSS of TEs/genes compared to tissue-specific eQTLs (two-sided Wilcoxon $p < 2.2e{-}16$) (Fig. 3b, Supplementary Fig. 8B). Additionally, we observed that shared eQTLs conserved their effect in both normal and tumor (Fig. 3c, Supplementary Fig. 8C). These results indicate that TE expression is under strong genetic control and that non-coding germline variants act as drivers of TE expression in cancer as similarly observed for gene expression[11].

## Transcription factors regulate TE expression more specifically in tumor

To corroborate the biological relevance of the discovered TE-eQTL variants we performed functional enrichment analysis of TE and gene eQTLs in normal and tumor using available ChIP-seq data from the Ensembl Regulatory Build[15] for 202 TFs and 29 histone marks. We then proceeded with multiple test correction with a given FDR of 5%

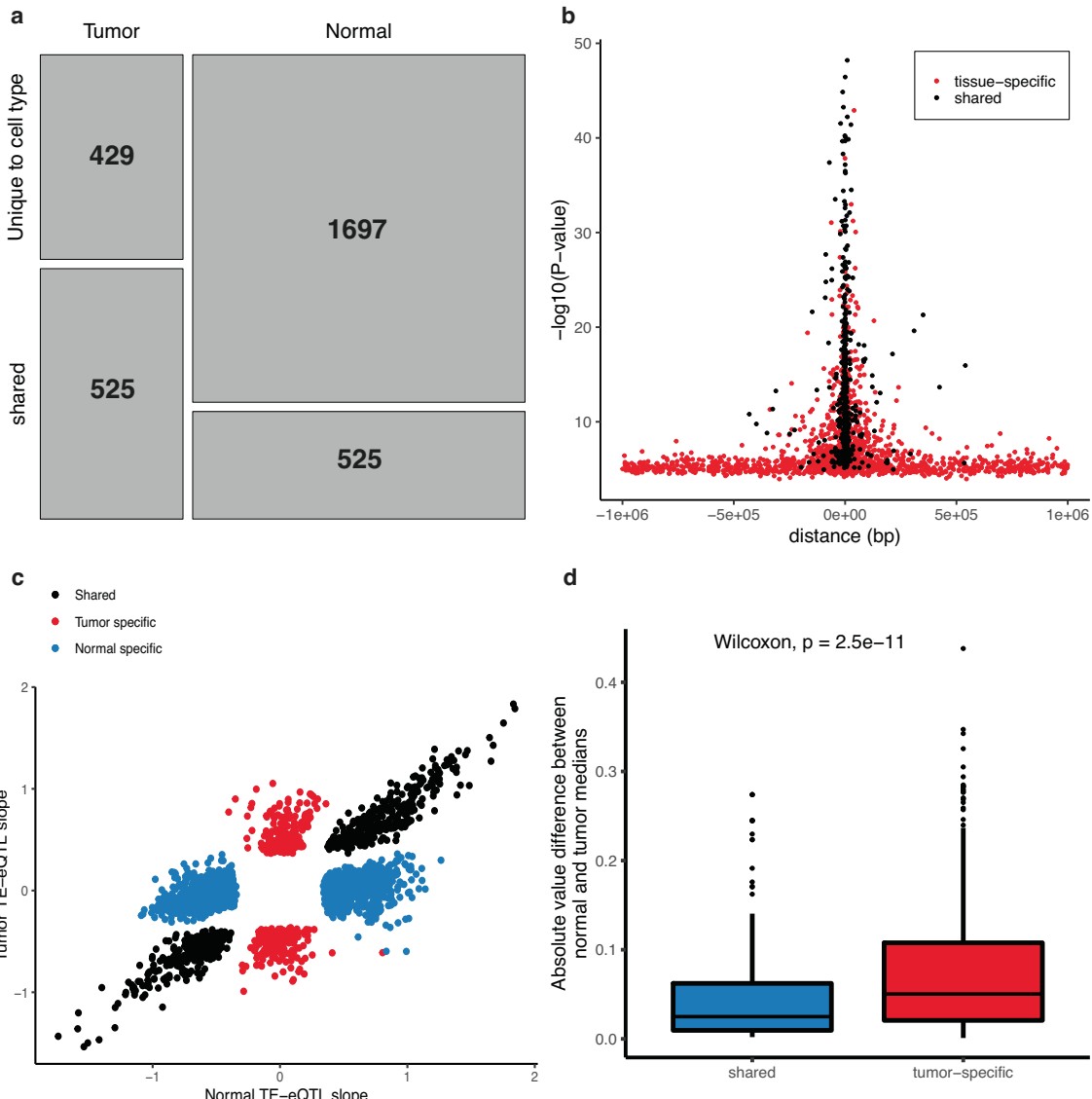

**Fig. 3 | Tissue specificity of TE-eQTLs. a** Mosaic plot of tissue specificity of TE-eQTLs. **b** Tissue specificity and distance of TE-eQTL to transcription start site (TSS). The shared TE-eQTLs (black) are closer to the TSS than are the tissue specific TE-eQTLs (red) (two-sided Wilcoxon $p < 2.2e-16$). **c** TE-eQTL slopes for the normal specific TE-eQTLs in blue, the tumor specific in red and shared in black. **d** Boxplot of the absolute value difference of median methylation betas between normal and tumor samples for shared ($n = 248$) and tumor-specific ($n = 675$) TE-eQTLs. We

observe significant increase in methylation change for tumor-specific TE-eQTLs compared to shared TE-eQTLs between normal and tumor (two-sided Wilcoxon test, $p = 2.5e-11$). Shared TE-eQTLs box plot values: minima = 1.987e−3; 1st quartile = 9.550e−3; median = 0.025; mean = 0.046; 3rd quartile = 0.062; maxima = 0.274. Tumor-specific TE-eQTLs box plot values: minima = 9.908e−4; 1st quartile = 0.021; median = 0.05; mean = 0.072; 3rd quartile = 0.11; maxima = 0.438. Source data are provided as a Source Data file.

(methods section 1.6.4). We found significant enrichment for many TF binding sites overlapping the eQTL loci highlighting the functional relevance of the variants discovered (Fig. 4a, b; Supplementary Figs. 9 and 10; Supplementary Data 8, 9). At 5% FDR, we discovered 5 significant hits (4 TFs and 1 histone marks) and 16 significant hits (12 TFs and 4 histone marks) that displayed stronger enrichment for TE eQTLs compared to gene eQTLs in normal and tumor, respectively. The TF most enriched over TE-eQTLs in normal tissues was ZNF274, a Krüppel-associated box (KRAB) domain-containing zinc-finger protein (KZFP), whereas the most enriched over tumor TE-eQTLs was TRIM28, the master corepressor that is recruited by the KRAB domain of many TE-binding KZFPs and serves as a scaffold for a heterochromatin-inducing complex capable of repressing TEs via histone H3 Lys9 trimethylation (H3K9me3), histone deacetylation and DNA methylation[16,17]. Additionally, BDP1 and BRF1, two subunits of the RNA polymerase III transcription initiation factor, were more enriched over TE-eQTLs

compared to gene eQTLs highlighting potential transcription of Alu or MIR TEs of the SINE family[18]. These results corroborate the biological relevance of TE eQTLs and point to possible transcription and repression of certain TEs.

To assess the differential effects of tumor-specific versus shared eQTLs, we performed functional enrichment analyses using available ChIP-seq data from LoVo colorectal cancer cells for 220 TFs and 2 histone marks[19] (methods section 1.6.4). We observed that in the case of genes, all tested TFs had a stronger enrichment for shared compared to tumor-specific eQTLs, indicating that these TFs are regulating gene expression in both the normal and tumor state. (Supplementary Fig. 11, Supplementary Data 10). In contrast, we found at 5% FDR, 60 significant hits (58 TFs and 2 histone marks) displaying stronger enrichment for tumor-specific versus shared TE-eQTLs, pointing to tumor-specific TE regulation (Fig. 4c; Supplementary Fig. 12; Supplementary Data 11). Of these, 23 were upregulated and 25 downregulated

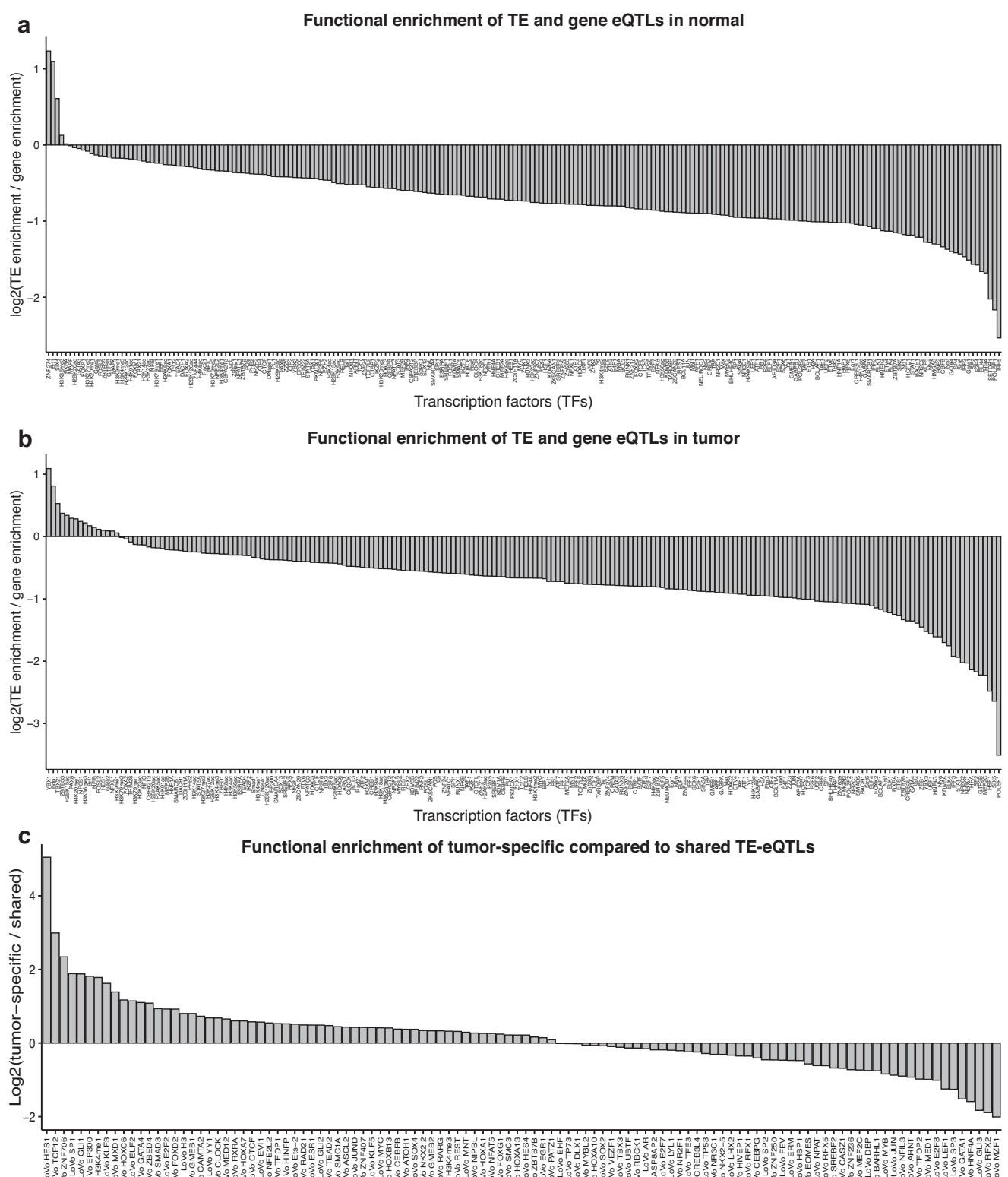

**Fig. 4 | Functional enrichment of eQTLs. a** The ratio between TE-eQTL enrichment and gene-eQTL enrichment in log2 scale discovered in normal. 5 TFs show stronger enrichment for TE-eQTLs in normal compared to gene-eQTLs. **b** The ratio between TE-eQTL enrichment and gene-eQTL enrichment in log2 scale discovered in tumor. We observed 16 TFs to have a stronger enrichment for TE-eQTLs than gene-eQTLs in normal. **c** log2 ratio between tumor-specific TE-eQTL enrichment and shared TE eQTL enrichment. We observe 60 TFs with a stronger enrichment for the tumor-specific TE-eQTLs than the shared eQTLs indicating that these TFs regulate TE expression specifically in tumor. Source data are provided as a Source Data file.

in tumors (12 were missing from our expression data and could not be tested for differential expression analysis), but we did not observe any significant correlation between the tumor-specific TE-eQTL enrichment to shared TE-eQTL enrichment ratio and fold change in the expression of the corresponding transcription factors (Pearson R = 0.019, *p*-value = 0.86; Supplementary Fig. 13). Thus, differential expression of these TFs is not driving the tumor-specific TE-eQTL effects. However, 61 of the 88 tumor-specific TE-eQTLs overlapping the binding sites of the 60 aforementioned TFs are not significantly associated (FDR = 5%) with any nearby (±1 Mb from TSS) TE or gene in normal, indicating that these regulatory regions are inactive in the normal state (Supplementary Fig. 14). Additionally, we compared methylation levels between normal and tumor samples for the tumor-specific and shared eQTLs and observed significantly increased (Wilcoxon rank sum test *p*-value = 2.5e$^{-11}$ for TEs and *p*-value = 0.0037 for genes) methylation over tumor-specific compared to shared eQTLs for both gene and TEs (Fig. 3d; Supplementary Fig. 8D).

Altogether these results suggest that many TFs are regulating TE expression. The inactivity of some of the TE eQTLs in normal and the significant changes in methylation between tumor-specific and shared TE-eQTLs indicate that regulatory switches involving the recruitment of these TFs might underlie the effects of tumor-specific TE eQTLs.

## Transposable elements as mediators of genetic effects onto genes

Having established that TEs are under genetic control, we next sought to assess the causal relationship between eQTL variants, TEs and genes and discover the extent to which TEs act as drivers of gene expression in tumor. To this end, we focused on regulatory variants affecting both TEs and genes and detected these in an unbiased manner by first associating TEs with genes using a similar approach to QTL mapping. Next, we quantified the identified 20,083 TE-gene pairs found in normal samples and 140,274 TE-gene pairs found in tumor at 1% FDR and used this quantified TE-gene pairs to find all eQTL-TE-gene triplets by performing a standard eQTL analysis (Methods; Supplementary Figs. 15–17). At 5% FDR, we discovered 11,937 and 9528 triplets in normal and tumor, respectively, for which we inferred the most likely causal relationship using Bayesian networks (Methods)[20–22]. We tested three models, (i) the causal model where the eQTL variant affects TE expression and then gene expression, (ii) the reactive model where the eQTL variant affects gene expression and then TE expression and (iii) the independent model where the eQTL variant affects independently TE and gene expression (Supplementary Fig. 18). Bayesian Networks were shown to be an adequate method for testing these three models[23]. We observed significantly more causal models in tumor (47%) compared to normal (23%) (Fisher *p*-value < 2e$^{-16}$) indicating that TEs are causal for gene expression predominantly in tumor and to a lesser extent in normal (Fig. 5a, b; Supplementary Fig. 19; Supplementary Data 12, 13). We also show that the proportion of causal models correlated with the genomic position of the TE with respect to the gene; intronic TEs tend to be reacting to gene expression. whereas TEs outside the gene body tended to be causal. We believe that the predominance of reactive model from intronic TEs and downstream of genes is a consequence of the transcription of the gene and not the transcription of the TE via the TE promoter. Interestingly, there were significantly more causal scenarios when the eQTL variant lied within the TE, rather than outside (Fisher *p*-value < 2e$^{-16}$) pinpointing to direct regulatory effects of the TE onto gene expression (Supplementary Fig. 20).

We then proceeded with replicating the causal inference of the eQTL−TE−gene triplets to corroborate our findings. We tested the triplets where all three molecular phenotypes were present in either GTEx colon transverse for the SYSCOL normal colon triplets or in TCGA-COAD for the SYSCOL tumor triplets, yielding 9577 (80%) triplets and 5893 (62%) triplets in common, respectively. We performed

BNs similarly to the original discoveries. We observe a high replication of both normal (62% similarity) and tumor (74% similarity) results (Supplementary Fig. 21; Supplementary Data 14, 15). We believe that the reason the replication of our normal colon eQTLs is lower than for the tumor eQTLs is because of sample size differences between SYSCOL normal colon (N = 275) and GTEx colon transverse (N = 174) decreasing our statistical power. These results corroborate our findings and highlight that our discoveries are valid.

Altogether, these results show that TEs are significantly more causal for changes in gene expression in tumor than in normal tissue.

## Transposable elements are drivers of gene expression during tumorigenesis

These results suggested that genetic variations in TE expression might drive tumorigenesis. To test this hypothesis, we considered the union of all triplets, i.e. the eQTL variant, TE and gene expression, discovered across tumor and normal tissue and using the same BN approach as previously mentioned, we inferred the causal relationship between the triplets in both states (methods). We similarly looked for shared triplets across the 11,937 normal and 9528 tumor triplets (eQTL-TE-gene triplets are the same in both states or the eQTL for TE-gene pair is in high LD (R² > = 0.9)). In both shared and union triplets, we observed a significant increase in the causal model in tumor (Fisher Exact Test *p*-value < 2.2e$^{-16}$ for shared and union triplets) mainly due to independent models and to a lesser extent reactive model shifting to causal. (Supplementary Fig. 22; Supplementary Data 16, 17). Focusing on the 9528 tumor triplets, we discovered 2584 (27%) triplets that switched to a causal model in tumor compared to normal, highlighting regulatory changes whereby TEs impacted the expression of nearby genes (Fig. 5c). These 2584 triplets constituted of 1766 unique TEs impacting 1558 unique genes. Interestingly, we observed that TEs switching to causal were significantly up-regulated compared to TEs that did not switch models between normal and tumor or that switched but not to causal (Wilcoxon *p*-value 2.2e$^{-14}$; Supplementary Fig. 23). These results suggest that upregulation of TEs could give rise to their gene expression driver capability.

While expression of most TEs was positively correlated with the expression of the associated gene in tumor (n = 2575) (Fig. 5d), only a few showed negative correlation (n = 9). Of the significant tumor TE-gene pairs tested in normal colon, we observed that 930 maintained the same effect (in terms of size and direction) whereas 36 showed an opposite effect in tumor samples. Interestingly, of the 1558 genes, 55 were cancer driver genes (CDG) (3 CRC specific; based on Cancer Gene Census[24]) but we did not find a significant enrichment of CDGs in triplets switching to causal compared to all other tumor triplets (Fisher exact test *p*-value = 0.2185; odds-ratio = 1.276). For 41 out of the 55 CDGs, we did not find a significant correlation between their expression and the expression of the corresponding TEs in normal samples pinpointing that these TEs have no impact on these genes in the normal state. Taken together, these results suggest an important role of TEs as drivers of gene expression during tumorigenesis.

## Non-coding germline variants activate driver TEs during tumorigenesis

We investigated whether any of the 9528 tumor triplets were constituted of any previously identified tumor-specific or shared TE-eQTLs and assess how the model likelihood changed between normal and tumor. We identified 320 and 133 tumor triplets constituted of a shared or a tumor-specific TE-eQTL, respectively (Fig. 6a, b) and observed that the 133 tumor triplets constituted with a tumor-specific TE-eQTL are significantly enriched for triplets switching to causal compared to the 320 tumor triplets constituted with a shared TE-eQTL (Fisher Exact test *p*-value = 6.6e$^{-4}$; Odds-ratio = 2.04) (Fig. 6b). Additionally, we observed that for 120 triplets with tumor-specific TE-eQTLs, the eQTL variant was not a significant eQTL for the corresponding gene in the triplet

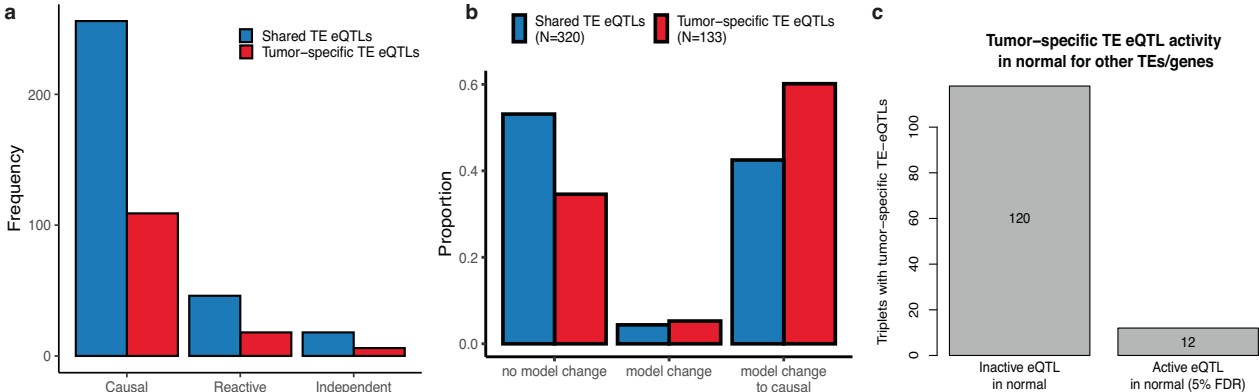

**Fig. 5 | Causal relationship between eQTL variants, TEs and genes. a** Barplot representing the mean probability for each of the three models in normal and tumor. We observe significantly more causal cases in tumor compared to normal (two-sided Wilcox *P*-value < 2e−16). **b** Barplot representing the model substitutions for the 9528 tumor triplets from normal to tumor. Independent models tend to shift to a causal in tumor. This is true also for the reactive models in normal but to a much smaller extent. **c** Barplot representing the number of triplets that do not switch models, that switch to a causal model or that switch to reactive/independent from normal to tumor. The majority of triplets do not switch models between normal and tumor. However, 2584 triplets are switching to a causal model making the corresponding TEs potential drivers of gene expression. **d** Each point represents a TE-gene for each of the 2584 tumor triplets. All points are significant in tumor but not in normal (gray points). We observe that in most cases, TEs are positively correlated with genes except for a few cases. Most cancer driver genes have no significant correlation with any TE in normal indicating that for most part, TEs impact them specifically in tumor. Source data are provided as a Source Data file.

**Fig. 6 | Tumor-specific and shared TE-eQTLs effects. a** The barplot represent the frequency of the causal, reactive and independent model for the triplets with shared or tumor-specific TE-eQTLs. **b** The barplot represents the model changes from normal to tumor for the triplets constituted of shared or tumor-specific TE-eQTLs. **c** Barplot that represent the number of tumor-specific TE-eQTLs that are inactive eQTLs for the triplet associated gene. Source data are provided as a Source Data file.

(Fig. 6c), highlighting that the eQTLs get activated in the tumor state influencing TE expression that subsequently impact gene expression. Altogether, these results suggest that tumor-specific TE-eQTLs contribute to tumorigenesis by impacting genes through TEs, adding additional proof that germline variants can be contributing to tumorigenesis.

### Driver TEs act as alternative promoters for genes in cancer

It has been shown that TEs could impact gene expression by acting as alternative promoters for nearby genes and creating chimeric transcripts (transpochimeric transcripts (tcGTs))[25,26]. To assess whether any of the tumor triplets with causal TEs were affected by tcGT events, we looked for cases where transcripts started from a TE and spliced into a single or multiple nearby genes (methods). We only kept tcGTs made up of the same TE and gene as in the 9528 tumor triplets and that were significantly more abundant in tumor samples compared to normal samples using a Fisher exact test. At 5% FDR, we discovered 126 tcGTs present in 147 tumor triplets. Of these, 78 triplets (66 TcGTs) were causal and 46 triplets (39 TcGTs) switched to causal from normal to tumor. Interestingly, we detected tcGT events with a known tumor suppressor gene RNF43 and two oncogenes ETS2 and SLCO1B3 supporting the extensive contribution of TEs during tumorigenesis.

## Discussion

Transposable elements are important contributors to tumorigenesis and provide supplementary means by which gene expression can be altered in cancer. While many studies have used a hypothesis-driven approach and focused at specific TEs or their subfamilies for discovering TEs that alter the expression of nearby genes in cancer[27–29], applying a genome-wide scan could allow to obtain a better picture of the effects of TEs on gene expression during tumorigenesis.

Here, we present a global profile of tumor drivers and show that TEs are highly prevalent mediators of genetic effects on genes altering their expression, specifically in tumor. By combining genome and transcriptome data together we, show that TEs are under tight genetic control and discover that transcription factors regulate TE expression much more in tumor than in normal. By looking at the interplay between eQTL variants, transposable elements, and gene expression, we are able to dissect eQTL effects and show that for several genes, the genetic effect of an eQTL is passed on genes through TEs which act as mediators and drive gene expression. We observe this to occur significantly more in cancer than in normal and show that the majority of TEs increase the expression of affected nearby genes. Interestingly, we discover that TEs affecting known cancer driver genes in cancer have for most part no significant effect on these genes in normal suggesting a tumor-specific effect of these TEs. Additionally, in our study we show that alongside predisposing alleles and somatic mutations, germline variants are crucial contributors to tumorigenesis as these allow for transcriptional changes to occur at the level of TEs that in turn result in altered expression of nearby genes in cancer as shown previously[11].

It is known that TEs are much more active in tumor than in normal, primarily due to a global hypomethylation in cancer driving their expression[26]. However, in our analysis we observe a higher number of eQTLs in normal than in tumor which may sound contradictory. This has to do with the nature of the tumor tissue being much more heterogeneous increasing the variance in the expression data subsequently affecting statistical power, leading to fewer eQTLs being discovered. By increasing sample size we could minimize this problem and increase the eQTL discovery in tumor.

To assess the function of these eQTLs, we used functional enrichment analysis. Even though we discover more eQTLs in normal, we observe that tumor TE-eQTLs are significantly enriched for more transcription factors compared to normal TE-eQTLs and the higher number of TFs significantly associated with tumor-specific TE-eQTLs indicates that TEs are more active in cancer compared to normal. Interestingly, we observed that for **61** tumor-specific TE-eQTLs, the eQTL loci is not an eQTL for any nearby gene or TE (±1 Mb) in the normal state. This indicates that these loci are probably inactive in normal and get activated during tumorigenesis, driving the expression of nearby TEs, specifically in cancer. Interestingly, we observed significant difference between DNA methylation changes at tumor-specific TE-eQTLs than shared TE-eQTL loci (eQTL active in both normal and cancer) which pinpoints that DNA methylation changes at these specific loci to be one of the causes of the activation of these eQTLs in cancer.

While we focused on TEs impacting the expression of nearby genes in an independent manner, it is highly plausible that synergistic effects occur from both cis- and trans- acting TEs. Performing such an analysis could give a fuller picture of the regulatory network behind the regulation of gene expression through TE effects, requiring, however, a high sample size for sufficient statistical power. Nevertheless, because of the highly repetitive nature of transposable element sequences and their evolutionary relatedness among TE families, mapping short reads originating from TEs is a real challenge[18,30]. Our RNA-seq dataset having a read length of 49 bp, it is highly possible that we did not map all expressed TEs subsequently leading to missing information, as shown previously[18,30]. Future studies where RNA-sequencing is performed with longer read lengths could allow for better mapping of expressed TEs and give us a fuller picture of the number of these driver TEs in cancer.

It is known that certain TE subfamilies, especially the younger ones like L1HS get reactivated during tumorigenesis and are able to retrotranspose creating tumor-specific integrations, perturbing the human genome. This is also a limitation in our study, as the assessment of tumor-specific TE integrations require Whole-Genome Sequencing (WGS) to be assessed. Additionally, our study is focused more on the effects of older TE subfamilies as these have accumulated sufficient mutations in their genomic sequence to make the various integrants distinguishable from each other. We believe that long-read sequencing technologies could be a good approach for studying younger TE subfamilies.

Altogether, we have outlined that TEs are important mediators of genetic effects onto genes that could potentially be used as risk factors or therapeutic targets for future drug development and aid in cancer treatment.

## Methods

### SYSCOL dataset

The Systems Biology of Colorectal cancer (SYSCOL) dataset (accession number: EGAC00001000204) contains data from genotypes and RNA-sequencing for matched normal-tumor samples (i.e., both tumor and normal samples originate from the same patient). Samples that had genotype data and molecular phenotype quantifications from tumor and normal (normal adjacent to tumor) tissue were analyzed, yielding 275 normal samples and 276 tumor samples. In case of multiple tumor samples from the same patient, only samples with quantifications from the most advanced tumor were kept.

### Genotypes

We used imputed genotypes and only kept variants with a minor allele frequency (MAF) ≥5%, yielding a total of 6,132,240 variants that were used for all downstream analyses.

### Transcriptome quantifications

**Read mapping.** SYSCOL samples were sequenced using 49 bp, 75 bp and 100 bp read lengths using paired-end non-stranded mRNA-sequencing. We first started by trimming all samples with 75 bp

($N = 73$) and 100 bp ($N = 4$) reads down to 49 bp to reduce any bias in downstream analysis stemming from read length. For this we used cutadapt[31] with the following command "cutadapt -u−Nreads -o <output_file><input_file>". All trimmed samples were mapped to the human reference genome (hg37) using hisat2[32].

**Transposable elements (TE) and genes quantifications.** Gene and transposable element counts were generated using the featureCounts software[33]. We provided a single annotation file in gtf format to featureCounts containing both genes and transposable elements. This prevents any read assignation ambiguity to occur. For transposable elements, we used an in-house curated version of the Repbase database[12] where we merged fragmented LTR and internal segments belonging to a single integrant. We only used uniquely mapped reads for gene and TE counts. Molecular phenotypes that did not have at least one sample with 20 reads and for which the sum of reads across all samples was lower than the number of samples, were discarded. Furthermore, we normalized molecular phenotypes (TEs and genes) for sequencing depth using the TMM methodology as implemented in the limma package of Bioconductor[34] and used gene counts as library size for both TEs and genes. Finally, we removed any molecular phenotype that had more than 50% of missing data (zeros) in tumor and normal samples separately and took the union of molecular phenotypes, yielding 17,430 genes and 50,921 TEs for a total of 68,351 molecular phenotypes.

**Normalization of molecular phenotypes.** The observed variability in molecular phenotypes from RNA-sequencing data can be of biological or technical origin. To correct for technical variability, while retaining biological variability, we residualised the molecular phenotype data for the covariates as described below:

1. To correct for population stratification that is observed between the SYSCOL samples, we used Principal Component analysis (PCA) results obtained from genotypes of SYSCOL patients. We only retained the first three principal components (PCs) as covariates.

2. In order to capture experimental/technical variability in the expression data, we performed PCA, centering and scaling, using pca mode from QTLtools software package[35]. To ascertain the number of PCs that capture technical variability, we used QTL mapping (see method 3.4.1 for the description of QTL mapping) to identify the best eQTL discovery power in both tumor and normal samples. To this end, we carried out multiple rounds of eQTL mapping for tumor and normal samples separately, each time using the 3 PCs from genotypes and incrementally adding 0, 1, 2, 5, 10, 20, 30, 40, 50, 60 and 70 PCs as covariates. This approach resulted in identifying 30 PCs in tumor and normal samples for maximizing eQTL discovery.

In total, 33 covariates were regressed out from tumor and normal sample expression data using QTLtools correct mode[35]. We additionally rank-normalized on a per phenotype basis across all samples such that quantifications followed normal distribution with mean 0 and standard deviation 1 N(0,1) using QTLtools --normal option[35].

**DNA methylation data and differential methylation of eQTLs**
We used microarray based DNA methylation data from the SYSCOL project, accession number EGAD00010001888, and a similar approach to a previous study to find differential methylation of eQTLs[11]. In brief, we calculated the absolute value difference of the medians of normalized methylation probe betas in normal and tumor that we call median differential methylation. We then compared the distribution of their medians in tumor-specific TE and genes eQTLs vs. the shared TE and gene eQTLs and calculated a P-value using the

Mann−Whitney U test. P-values were corrected for multiple testing using the R/qvalue package with a given FDR threshold of 5%.

**Differential TE/gene expression analysis**
The DESeq2 R package[36] was used in calculating differentially expressed genes and TEs. We normalized the raw TE/gene counts within the DESeq2 package using the sequencing date, GC mean and insert size as covariates. The differential expression P-values were corrected for multiple testing using an FDR threshold of 5%.

**Transcriptome QTL analysis**
All analyses were performed separately for normal and tumor samples. We used imputed genotypes with MAF ≥ 5%, gene expression data with normalized counts per million (CPMs) (as described above) for both eQTL and conditional eQTL mapping.

**Expression quantitative trait loci (eQTL) mapping.** For eQTL mapping, we used cis mode of the QTLtools software package[35]. For each molecular phenotype:

1. We counted all genetic variants in a 1 Mb window (+/−1 Mb) around the transcription start site (TSS) of the phenotype and tested all variants within this window for association with the phenotype. We only retained the best hits which are defined as the ones with the smallest nominal p-value.

2. Next, we used permutations to adjust the nominal p-values for the number of variants tested. More specifically, we randomly shuffled the quantifications of the phenotypes 1000 times and retained only the most significant associations. This created a null distribution of 1000 null p-values. Then, we fitted a beta distribution on the null distribution and used the resulting beta distribution to adjust the nominal p-value. Principally, this strategy allows to quantify the chance of getting a smaller p-value than the nominal one in random datasets.

This effectively gave us the best variant in cis together with the corresponding adjusted p-value of association for each molecular phenotype. Finally, to correct for the number of phenotypes being tested we used False Discovery Rate (FDR) correction approach. More specifically, we used the R/qvalue package[37] to perform genome-wide FDR correction which ultimately facilitated to extract all phenotype-variant pairs that are significant at a pre-determined FDR threshold, 5% FDR in this case.

**Conditional analysis for eQTL mapping.** The cis mode informs us on the best phenotype-variant pair only. Given that the expression of molecular phenotypes can be affected by multiple cis eQTLs, we proceeded with conditional analysis to discover all eQTLs with independent functional effects on a given phenotype. Principally, discoveries are made after conditioning on previous ones. Again, cis mode in the QTLtools software package was used[35]. In brief, after running permutations (method 1.4.1) for each phenotype, we calculated a nominal p-value threshold of being significant. We first determined the adjusted p-value threshold that corresponds to the targeted FDR level and then used the beta quantile function to go from adjusted p-value to a specific nominal p-value threshold. For conditional analysis, forward-backward methodology is used to discover all independent QTLs and to identify the most likely candidate variants, while at the same time controlling for a given FDR (5% FDR in this case). We only kept the top variant for each signal.

**Tissue-specific and shared eQTL analysis.** To discover tissue specific and shared eQTLs, we used the eQTL results obtained after running the conditional pass. In total, we tested 17,186 eQTLs to discover normal-specific eQTLs and 6751 to discover tumor-specific eQTLs. To do that,

we used linear mix models using an interaction term between dosage and tissue (i.e tumor or normal) to test whether the slopes in normal and tumor are significantly different. Linear mix models are needed here because normal and tumor samples are originating from the same patient thus genotypes will be identical. We did this for tumor and normal eQTLs separately. Then we performed multiple test correction using the R/qvalue package[37] with a given FDR threshold of 5%. Additionally, for all significant results at 5% FDR, if eQTL slopes (slopes given from conditional QTL mapping using QTLtools) in normal and tumor had the same direction, then we only kept the ones where the SNP-phenotype association in the opposite tissue was not nominally significant ($P > 0.05$) as given by the cis nominal pass mode in the QTLtools package[35].

Shared eQTLs are defined as the ones where the $P$-value for the interaction term is not significant but need to be significant eQTLs (5% FDR) in both normal and tumor as assessed by the conditional QTL mapping.

**Functional enrichment analysis.** To compare the QTL variants to a null distribution of similar variants without regulatory association, we sampled for each eQTL variant 100 random regulatory genetic variants matching for relative distance to TSS (withing 2.5 kb) and minor allele frequency (within 2%) and only kept variants that are not eQTLs for any other TE or gene (nominal $p$-value > 0.05). The enrichment for a given category was calculated as the proportion between the number of regulatory associations in a given category and all regulatory variants over the same proportion in the null distribution of variants. The $p$-value for this enrichment is calculated with the Fisher exact test. Finally, we corrected for multiple testing using an FDR threshold of 5% using the "p.adjust" function in the R programming language. The code for performing the functional enrichment analysis can be accessed here: https://github.com/NLykoskoufis/fenrichcpp[38].

**Ensembl regulatory build ChIP-seq dataset.** ChIP-seq data was downloaded from the FTP site (http://ftp.ensembl.org/pub/grch37/current/regulation/homo_sapiens/). The dataset contains ChIP-seq data from 88 human cell types for a total of 209 transcription factors and 29 histone marks (build hg19). For each of the TFs and histone marks, we took the union of all peaks together from all 88 cell types. Overlapping peaks were merged together using the "merge" options in the BEDtools software[39]. This allowed us to create an extensive annotation of peaks for 209 TFs and 29 histones genome-wide.

**Colorectal cancer LoVo cell line ChIP-seq dataset.** We used publicly available ChIP-seq data from colorectal cancer LoVo cell line with accession code GSE49402. The dataset comprises of 202 TFs and 2 histone marks (build hg19). We used BED files containing the coordinates of the peaks for each TF and histone mark for functional enrichment of our eQTLs.

For gene and TE eQTLs in normal and tumor, we used the peak annotation generated from the Ensembl Regulatory Buil data to get an extensive comprehension of which TFs regulate the expression of TEs. Regarding the tumor-specific vs. shared TE and gene eQTLs, we used available ChIP-seq data from the colorectal cancer LoVo cell line[19]. We used a cancer specific dataset as we were interested in discovering cancer specific effects.

**Testing for associations between TEs and genes**
To discover associations between TEs and genes, we proceeded in a similar way to what we did for QTL mapping (method 1.4.1). Effectively, we used TE expression as our "genotypes" and genes as our phenotype. Then, we corrected for multiple testing using the R/qvalue package with a given FDR of 1%. We then estimated the nominal $p$-value thresholds for each phenotype being tested as described in (method 1.4.2) with a given FDR of 1%. Given the nominal threshold we get for

each gene, we then extracted all TEs with an association $P$-value below this threshold which could give multiple TEs for a gene in some cases.

**Quantifying TE-gene pairs**
To quantify each of TE-gene pairs that have been found to be significant, we used a dimensionality reduction approach based on PCA as previously described[22]. Specifically, for each TE-gene pair, we aggregated gene expression together with TE expression by using the coordinates on the first principal component. This effectively built a quantification matrix with rows and columns corresponding to the number of TE-Gene pairs and individuals, respectively. All quantifications have been rank-normalized on a per phenotype basis so that the values match a normal distribution N(0,1). This prevents outlier effects in downstream association testing. This is all implemented in the clomics software package[22].

**Causal inference by Bayesian networks for QTL-TE-gene triplets**
Bayesian networks (BNs) are a type of probabilistic graphical model that uses Bayesian inference to compute probabilities. BNs aim to model conditional dependencies and therefore causation by representing conditional dependencies as edges and random variables as nodes in a directed acyclic graph. The flow of information between two nodes is reflected by the direction of the edges, giving an idea of their causal relationship. BNs have been previously used in a genetic framework[20] to get insight into the most likely network from which the observed data originates.

In BNs, the joint probability density can be divided into marginal probability functions and conditional probability functions for the nodes and edges, respectively. Additionally, BNs satisfy the local Markov property where each variable is conditionally independent of its non-descendants given its parent variables. In the context of this study, we used BNs to learn the causal relationships between triplets of variables, each one containing a genetic variant, a transposable element and a gene. In practice, only three distinct network topologies where relevant to the hypotheses we wanted to test (Supplementary Fig. 12). More specifically, we looked at:

1.  The causal scenario where the genetic variant affects first the TE and then the gene.
2.  The reactive scenario where the genetic variant affects the gene first and then the TE.
3.  The independent scenario in which the variant affects the gene and the TE independently.

Of note, we only retained network topologies that assume that the signal systematically originates from the genetic variant. In practice, we applied BNs on data that was obtained from running an QTL mapping using the TE-gene pairs using a similar approach to QTL mapping described above (Method 1.4.1) and only kept significant results at 5% FDR which corresponds to 11,937 QTL-TE-gene triplets in normal and 9528 QTL-TE-gene triplets in tumor.

For each triplet, we build a 275 ×3 matrix in normal and 276 ×3 matrix in tumor containing normalized quantifications and used it to compute the likelihood of the 3 BN topologies using the R/bnlearn package (Version 4.5)[40]. As a last step, we went from likelihoods to posterior probabilities by assuming a uniform prior probability on the three possible topologies. Posterior probabilities where used for all BN-related analyses.

**Transpochimeric transcripts analysis**
First, a per sample transcriptome was computed from the RNA-seq bam file using StringTie[41] with parameters −j 1 −c 1. Each transcriptome was then crossed using BEDTools[39] to both the ensembl hg19 coding exons and curated RepBase[12] to extract TcGTs for each sample. Second, a custom python program was used to annotate and aggregate the sample level TcGTs into counts per groups (normal, tumor). In

brief, for each dataset, a GTF containing all annotated TcGTs was created and TcGTs having their first exon overlapping an annotated gene or TSS not overlapping a TE were discarded. From this filtered file, TcGTs associated with the same gene and having a TSS 100 bp within each other were aggregated. Finally, for each aggregate, its occurrence per group was computed.

### GTEx dataset

We downloaded available data for colon transverse ($N = 174$) and germline genotypes from dbGAP (accession code: phs000424.v8.p2).

**Germline genotypes.** We used the already filtered VCF file provided by GTEx. The following filters were applied and kept all variants with a MAF $\geq 5\%$, yielding a total of 6,494,417 variants.

**RNA-seq dataset.** The RNA-seq dataset was treated similarly to SYSCOL RNA-seq data. We first trimmed the reads down to 49 bp using cutadapt[31]. Then we mapped and quantified the samples using the exact same approach as for SYSCOL (methods section 1.3.2). Finally, we combined all samples together into a multi-sample bed file and kept all features (TEs, genes) that had less than 50% of missing expression data across all samples, yielding a total cd of 167,429 TEs and 18,472 genes. Then, we corrected our expression data using the first 3 principal components (PCs) obtained from genotypes, the sex of the samples, the platform they were sequenced and the first 20 PCs obtained from the expression data, for a total of 25 covariates used.

### TCGA dataset

We downloaded available germline genotypes and RNA-seq data for colon adenocarcinoma ($N = 251$) from The Cancer Genome Atlas (TCGA) database, accession code phs000178.v11.p8.

**Germline genotypes.** Germline genotypes were downloaded from the legacy archive GDC portal. We downloaded all germline genotypes for TCGA-COAD in birdseed format. We used birdseed2vcf python tool (https://github.com/ding-lab/birdseed2vcf) to convert birdseed to VCF format. We then combined all samples together creating a multi-sample VCF file that we spitted per chromosome and uploaded to the Michigan Imputation Server[42] for imputation and phasing using the Haplotype Reference Consortium (HRC) as reference panel, Eagle v2.4 software[43] for phasing and European (EUR) population. Finally, we merged all chromosome VCFs into a single VCF file and kept variants with a MAF $\geq 5\%$, HWE $> 1e{-}06$ and $R^2 > 0.3$, yielding a total of 5,511,779 variants.

**RNA-seq data.** As the read length of TCGA-COAD samples is the same as SYSCOL, we did not need to trim the reads. We mapped, quantified, and filtered our RNA-seq data in a similar way as for SYSCOL and GTEx colon transverse samples yielding a total of 19,376 genes and 75,815 TEs. Expression data was corrected using the same approach as for SYSCOL (methods section 1.3.3) using the first 3 principal components (PC) obtained from genotypes and the first PC obtained from expression data for a total of 4 covariates used.

### Replication of eQTL findings

For the replication of our normal and tumor eQTL discoveries, we used the "rep" mode in the QTLtools software[44]. We then used the pi1 metric to estimate the proportion of significance of our eQTLs in GTEx colon transverse. The pi1 is equal to 1−pi0 where pi0 is the proportion of true null $p$-values obtained using pi0est from the Qvalue R package[45].

### Replication of the eQTL−TE−gene triplets

We used the same eQTL−TE−gene triplets discovered in normal and tumor and replicated them in GTEx or TCGA-COAD, respectively. We used the exact same approach as previously (methods section 1.8). We

then calculate the mean probability of the causal, reactive and independent model. Finally, we compared the percentage of triplets with the same model predicted in both SYSCOL and the replication dataset.

### Reporting summary

Further information on research design is available in the Nature Portfolio Reporting Summary linked to this article.

## Data availability

All data generated during the current study are available in Supplementary Data 1–17. Supplementary Data 1–17 contains results obtained from the various analysis performed using raw publicly available datasets. The RNA-sequencing data and genotype arrays for the 275 normal colon and 276 colorectal cancer samples from SYSCOL is available in the European Genome-Phenome Archive (EGA) under accession code EGAC00001000204. This restricted data can be requested through EGA or cla@ki.au.dk. Questions regarding the processed data can be emailed to Dr. Nikolaos Lykoskoufis at nikolaos.lykoskoufis@gmail.com. The microarray based DNA methylation data from the SYSCOL project is available in EGA under accession code EGAD00010001888. The RNA-sequencing and germline genotypes from the GTEx dataset can be obtained through dbGAP under accession code phs000424.v8.p2. The RNA-sequencing and germline genotypes from TCGA database can be obtained through dbGAP under accession code phs000178.v11.p8. Both GTEx and TCGA datasets are restricted data and access can be requested through dbGAP. Colorectal cancer LoVo cell line ChIP-seq data can be obtained from Gene Expression Omnibus (GEO) under accession code GSE49402. Transcription factors and histone marks ChIP-seq data can be downloaded from the Ensembl FTP site [http://ftp.ensembl.org/pub/grch37/release-100/regulation/homo_sapiens/Peaks/] where we downloaded all compressed bed files for all cell types. The full list of hyperlinks of all ChIP-seq datasets from Ensembl used in the current study can be found in Supplementary Dataset 18. Source data are provided with this paper.

## Code availability

All custom scripts used can be accessed here: https://github.com/NLykoskoufis/te_project. The code for the functional enrichment analysis can be accessed here: https://github.com/NLykoskoufis/fenrichcpp[38].

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

## Acknowledgements

The computations were performed at the University of Geneva on the Baobab cluster. This work was supported by grants from Louis-Jeantet Foundation support (to E.T.D.) and SNSF grant (to E.T.D.). This work was supported by grants from the European Research Council (KRABnKAP, No. 268721; Transpos-X, No. 694658), the Personalized Health and Related Technologies (PHRT-508) program, the Swiss National Science Foundation (310030_152879 and 310030B_173337), the Swiss Cancer League and the Aclon Foundation to D.T. The funders had no role in study design, data collection and analysis, decision to publish, or preparation of the manuscript.

## Author contributions

N.M.R.L., H.O. and E.T.D. designed the study. N.M.R.L. analyzed the data and wrote the manuscript and N.M.R.L., H.O., D.T. and E.T.D. interpreted the results. E.P. shared the quantifications data. H.O., D.T. and E.T.D. jointly supervised this work.

## Competing interests

E.T.D. is currently an employee of GSK. The work presented in this manuscript was performed before he joined GSK. N.M.R.L. is currently an employee of NGS-AI JSR Life Sciences. The work presented in this manuscript was performed before he joined NGS-AI. All other authors declare no competing interests.
