## [Peer Review File · Nature Communications]

Editorial Note: Figures in this Peer Review File has been amended to remove third-party material where no permission to publish could be obtained.

REVIEWER COMMENTS

Reviewer #1 (Remarks to the Author): Expert in cancer genetics, genomics, eQTLs

In this reviewed manuscript, the authors analyzed genome-wide genotype information with transcriptomic profile across 275 normal and 276 colorectal cancer samples from SYSCOL cohort. They mapped transposable (TE)-eQTLs in these samples separately. Using a Bayesian network, they found TE can play as mediators of genetic effect and alter strong nearby gene expression in tumor than normal. The manuscript reports a systematic TE-eQTL analysis in colorectal cancers. The idea of using eQTL analysis to investigate genome-wide genetic regulation of TE is interesting, and the authors followed a clear logic to organize this manuscript, although many findings are expected. In addition, there are many grammatical mistakes and typographical errors throughout the manuscript. More importantly, several noteworthy significant limitations exist. The manuscript will need a throughout revision before further consideration.

1. The authors discarded repetitive reads and applied a simple quantification method for TE quantification. This approach has been extensively proved as problematic since TEs contain highly repeated sequences. Also, considering the sample reads are only 49bp, few TEs can likely be uniquely mappable (PMID: 32576954). Thus it is hard to evaluate the reliability of TE quantification. Many bioinformatics approaches developed for TE quantification, such as REdiscoverTE (PMID: 31745090), consider multi-mapped reads and are evidenced as more reliable. The authors should benchmark the accuracy of their methods in TE quantification.
2. The authors used the Bayesian networks model for causal inference methods, while there are several other more widely used, such as Mendelian randomization (PMID: 29164242), LCV (PMID: 30374074), and SEM (PMID: 29447406). The authors should discuss the advantages of Bayesian networks over others.
3. Figure 2A-B, the positional distribution is hard to reveal the more proximal of TSS-eQTL than gene-eQTL, the authors may need to clarify the differences using a new figure.
4. Figure 4. The authors found 5, 15 TFs that show more robust enrichment of TE-eQTL than gene-eQTL, respectively. However, the number cannot be reflected in figure 4. Why there are 7 and 20 enriched TFs in their figure A and B?
5. For example, in Methods section 1.3.1, the authors trimmed reads to 49 bp. They should describe how the trimming was performed.
6. The authors should explain how to transform TE expression level to "genotype" data.

7. In Methods section 1.6.4, the authors should explicitly describe the CHIP-seq data sets used.
8. The authors should explicitly list the source of their genotype and RNA-seq data and how to access these data.
9. Figure 1A. please check the axis label.
10. Figure 6. The legend does not match the figure. Where is figure 6D?
11. The authors should host the analysis code for reproduction in a publicly accessible repository such as Github and the TE-eQTLs in a public source that can be downloaded.

Reviewer #2 (Remarks to the Author): Expert in colorectal cancer genetics, genomics, eQTLs

This work analyzes the role of transposable elements in gene expression regulation, taking into account genetic variation and DNA methylation.

The analyses show interesting global associations, with long lists of genes that might be regulated by TEs. The role in cancer, however, is not well established, since these are observational associations and the Bayesian networks used to infer causality just rely on the strength of the statistical associations.

Only one dataset has been used, with no attempt to replicate at least the results related to tumor tissue in the larger TCGA dataset.

The normal tissue used in the study was obtained from cancer patients, and its gene expression might be altered by the presence of tumor by diverse mechanisms. Again, no attempt has been made to validate the results in other datasets of normal colon tissue.

Many of the methods used involve in-house curated databases, which are not made available and difficult the possibility to replicate the results.

The statistical significance of the results are only mentioned in the methods section and, though the methods seem reasonable, reading the results often brings the question "has this been adjusted for multiple comparisons, and how?"

As example, only 5 TFs in normal and 15 in tumor were enriched for TE. How many TFs were analyzed? What were the p-values? Authors refer to supplementary tables 6 & 7, but the reported genes in the

text are within a long list of significant and non-significant results, and transcription factors are not marked. Supplementary tables don't have a legend to clearly explain their contents.

The functional enrichment is based on matching ChIP-seq peaks of Ensemble Regulatory Build. All cells are combined for the analysis of eQTLs and TEs, but only cancer LoVo cell line for the comparison of tumor vs shared. Authors could discuss how the diverse cell types combined might affect the functional enrichment results, and the possibility of analyzing ChIP-seq experiments of normal colon cells.

The discussion sentences are difficult to match to specific results. For example, there is emphasis of findings more relevant in tumors than in normal, but there are 10111 TE-eQTL in normal and 5152 in tumor, a similar ratio for gene-eQTLs. Only enrichment of transcription factors are more often in tumor than normal, but only 15 and 5 were found, respectively.

Reviewer #3 (Remarks to the Author): Expert in transposable elements and genomics

The authors provide a bioinformatic analysis, which leads to the hypothesis that tumor-specific TE-eQTLs modify the expression of TEs and subsequently alter the expression of an important number of genes in colorectal cancer, including cancer driver genes. The authors also build evidences for the relation of eQTLs, TEs and genes in the regulatory switches occurring from healthy tissue to cancer. The paper is well written, and the emergent hypothesis will be of great interest to the Transposable Elements community.

Below, I have noted some comments for the authors to consider when revising their paper for publication in Nature Communications.

Comments

Page 4 line76-77. As the authors know, filtering for 'uniquely mapped reads' very challenging for young TE families (as explained by Lanciano and Cristofari. Nat Reviews 2020 cited in the bibliography).

- when you find expression of a TE in a donor and not in another due to uniquely mapped reads, how do you know that the donor with no expression simply doesn't have a different SNP variant that you are using to track the uniquely mapped read? And when you see more expression in a tumor sample than in a healthy matched tissue, how do you know that this is not due to the TE increasing its copy number in tumor samples (active ones through transposition, and older ones through

rearrangements/duplications)? I consider that the authors should validate their results with unique reads mapping the junctions TE-genome for the younger TEs as the bona fide way to uniquely map the expression of a specific TE locus (particularly in the case of young L1s).

-Fig 1.A. It would be really informative to analyse the proportion of TE subfamilies that are expressed in tumor and normal tissue in different graphs to estimate the differences in expression in both types of tissues.

Page 5 line 102. The authors claim that the smaller number of eQTL per TE compared to genes could be due to smaller evolutionary time of TE regulatory landscapes. The authors could test this hypothesis calculating the number of eQTL for TEs groups that contain a succession of subfamilies with a wide and well established age scale (LINEs for example).

Page 7 line 152. The identification TFs involved in the expression of tumor-specific eQTLs has been done with CHIP-seq data from LoVo colorectal cancer and the authors suggest that changes in methylation patterns allow more accessibility of those TFs in tumor. To increase the robustness of the proposed mechanism, a CHIP-qPCR in colorectal cancer and primary colon cell lines would be highly recommendable to validate these results. Another, no-wet lab option to demonstrate that the TE-eQTLs are more accessible to TFs will be the comparison with ATAC-seq data from colorectal cancer/normal tissues.

Page 8 line 185. The authors detect that the "reactive model" is more frequently for intronic TEs. The authors have not elaborated if this could be a consequence of the TE being transcribed within the gene rather than as a consequence of the TE promoter stimulation. It is expected a 50% distribution of sense/antisense TE insertion in Introns (and for LINE-1s, this appears to be even bias towards antisense), so it could be interesting to select antisense intronic TEs and detect if the TE reads in these cases are in sense respect to the gene or in sense respect to the TE (if RNA-seq data has been prepared with stranded-kits to detect transcriptional direction).

Page 10 line 238. It appears that the tumor specific TE-eQTLs contribute to tumorigenesis by impacting genes through TEs. In fact, the authors shown that the expression of 34 CDGs was positively correlated with the expression of TEs exclusively in tumor. Ideally, experimental validation would be necessary for this kind of associations. Others authors have already CRISPR deleted TEs (Chuong, Elde and Feschotte, Science 2016). Authors could remove the TE in cell lines and verify the succession of events that they are proposing here.

Manuscript NCOMMS-21-48187 “Transposable elements mediate genetic effects altering the expression of nearby genes in colorectal cancer”

Point by point responses to reviewers

Reviewer #1: In this reviewed manuscript, the authors analyzed genome-wide genotype information with transcriptomic profile across 275 normal and 276 colorectal cancer samples from SYSCOL cohort. They mapped transposable (TE)-eQTLs in these samples separately. Using a Bayesian network, they found TE can play as mediators of genetic effect and alter strong nearby gene expression in tumor than normal. The manuscript reports a systematic TE-eQTL analysis in colorectal cancers. The idea of using eQTL analysis to investigate genome-wide genetic regulation of TE is interesting, and the authors followed a clear logic to organize this manuscript, although many findings are expected. In addition, there are many grammatical mistakes and typographical errors throughout the manuscript. More importantly, several noteworthy significant limitations exist. The manuscript will need a throughout revision before further consideration.

Point 1: The authors discarded repetitive reads and applied a simple quantification method for TE quantification. This approach has been extensively proved as problematic since TEs contain highly repeated sequences. Also, considering the sample reads are only 49bp, few TEs can likely be uniquely mappable (PMID: 32576954). Thus it is hard to evaluate the reliability of TE quantification. Many bioinformatics approaches developed for TE quantification, such as RDiscoverTE (PMID: 31745090), consider multi-mapped reads and are evidenced as more reliable. The authors should benchmark the accuracy of their methods in TE quantification.

The SYSCOL mRNA sequencing dataset was sequenced in paired-end configuration. Even with a read length of 49 base pairs (bp), the majority of expressed transposable elements (TEs) are uniquely mappable. We assessed whether multi-mapped reads increase the number of aligned reads and whether they add additional information. Specifically, we used featureCounts [1] to quantify SYSCOL samples using uniquely mapped reads only or taking multi-mapped reads (using the “-M fraction” option) into account. We then estimated the proportion of unique reads mapping to intergenic TEs compared to multi-mapped reads. We observed that most samples have a very high proportion of uniquely mapped reads on TEs (more than 90%) and the multi-mapped reads are in the minority (<10%) indicating that using uniquely mapped reads gives a robust and reliable quantification profile of TE expression and that multi-mapped reads do not improve read alignment to a greater extent (**Figure 1**).

Point 1 – Figure 1: Figure representing the proportion of uniquely mapped reads (black) and multi-mapped reads (grey) of intergenic TEs for all 551 SYSCOL samples. We observe that the majority of reads are uniquely mapped with only a minority being multi-mapped reads.

In Sexton et al., 2019 [2], the authors explored the mappability of transposable elements in the human genome. First, they compared single-end and paired-end TE mappability using reads of 76bp length. We observe a clear trend for an increase in the mappability score with paired-end alignment (**Figure 2**).

Point 1 – Figure 2: From Sexton et al., 2019 “Comparison of single-end and paired-end TE mappability. Each point represents 1 TE sequence. Paired-end mappability scores were generated from 76 base pair read alignments with ≤ 3 mismatches. Single-end mappability

scores were generated from GEM Mappability ($k = 76$) with the default of a 4% mismatch rate. Mappability scores were aggregated for each TE as the percent of base pairs in the sequence with a unique mappability score.” [2]

Secondly, Sexton et al., 2019 assessed the general mappability of TE sequences. Figure 3 outlines the percentage of loci in each subfamily (colored by family) that are uniquely mappable for different read lengths either in single-end or paired-end alignments. We observe that the highest percentage of uniquely mappable loci are achieved with 100bp fragments in paired-end configuration. Importantly, a high number of TEs are uniquely mappable also with 50bp paired-end configuration. Sexton et al., 2019 estimated that using read lengths of 50bp in paired-end configuration allows to uniquely map around 68% of transposable elements in the human genome [2]. These data provide confidence that using a read length of 50bp in paired-end configuration is sufficient for accurately uniquely mapping locus-specific transposable elements. However, using longer read lengths of 100bp in paired-end configuration would allow us to obtain a more complete picture of the TE transcriptome.

Point 1 – Figure 3 (From Sexton et al., 2019): “Percentage of mappable elements for each family and subfamily of TEs. Colored by class distinction, each bar represents the percentage of unique elements in each subfamily. An element is only considered unique if every position in its sequence has a unique mappability score. Both GEM Mappability and Bowtie were run with an allowance of 3 mismatches” [2].

Moreover, TE expression at single loci level is needed for eQTL analysis. Unfortunately, RediscoverTE [3] does not provide estimates for this. It rather provides expression at TE subfamily level. Other tools, e.g., Tetranscripts [4] do provide TE expression quantification at single loci level and assign multimapped reads using an expectation-maximization (EM) algorithm. In practice multimapped reads are assigned to the TE loci candidates that have more uniquely mapped reads and the distribution of uniquely mapped reads is used as a likelihood for multimapped read assignation. Whereas these tools claim they increase the number of aligned reads, we believe the increase is minimal on top of 50bp paired-end sequencing. Consequently, we opted for a conservative approach and considered only uniquely mapped reads as evidence for expression, as opposed to using the EM-based guess that is biased towards TE loci having a lesser repetitive sequence.

Point 2: The authors used the Bayesian networks model for causal inference methods, while there are several other more widely used, such as Mendelian randomization (PMID: 29164242), LCV (PMID: 30374074), and SEM (PMID: 29447406). The authors should discuss the advantages of Bayesian networks over others.

This is a very good point made by the reviewer as indeed there are many different causal inference methods available, i.e. mendelian randomization (MR), structural equation modelling (SEM), latent causal variable (LCV) models and many others.

In 2017, Ainsworth et al., [5] published a study where they compared various methods for inferring causal relationships (MR, BN and SEM were among them) between genotype and phenotype using additional biological measurements. They created 12 possible causal models explaining the relationship between a genetic variant G and two observed traits X and Y (**Figure 1**). As we can see, models (a), (b) and (c) are the three models we also tested in our project where the (a) would be the independent model, (b) the causal and (c) the reactive model.

[Redacted]

For each of the 12 causal scenarios given in the figure above, they simulated 1,000 replicated datasets containing 1,000 individuals. Next, for each simulated data set they applied the various causal inference methods and assessed how well these methods could recover the true causal structure. Figure 2 (modified from Ainsworth et al., 2017 [5]) shows the results of the simulations using BNs (R/bnlearn package), SEM and MR.

[Redacted]

From the figure above, we observe that Bayesian networks using the R/bnlearn package [6] and SEM successfully managed to predict the correct model from the simulated data for models (a), (b) and (c) (which we are the most interested in) [5]. However, we observe that MR were unable to properly predict models (a) and (c). This has to do with the different assumptions that need to be fulfilled for MR to be used. In our scenario, the genetic variant G must be associated with X (TE or gene) and associated with Y (gene or TE) only via its association with X [7, 8], which means that our independent model ($G \rightarrow X; G \rightarrow Y$) would violate this assumption. Furthermore, MR assumes that a genetic variant considered as an instrumental variable affects the outcome only via the exposure variable/ risk factor tested.

In our case, we observed genetic variants to be significantly associated with multiple genes and multiple TEs, thus violating the above-mentioned assumption.

These results suggest that using BNs or SEM can be used to correctly predict causal inference of the underlying models (a), (b) and (c). We decided to go forward with the BN approach described in our study as it has already been used in previous published papers [9-11] indicating that the approach is valid and robust.

Point 3: Figure 2A-B, the positional distribution is hard to reveal the more proximal of TSS-eQTL than gene-eQTL, the authors may need to clarify the differences using a new figure.

We agree with Reviewer #1 that the positional distribution of proximal TSS-eQTLs and gene-eQTLs on Figure 2A-B is not clearly distinguishable. For better visualization we had added a Supplementary Figure 5 in the original manuscript to clearly show the the difference in distance from TSS between TE-eQTLs and gene-eQTLs in both normal and in tumor.

Point 4: Figure 4. The authors found 5, 15 TFs that show more robust enrichment of TE-eQTL than gene-eQTL, respectively. However, the number cannot be reflected in figure 4. Why there are 7 and 20 enriched TFs in their figure A and B?

We thank the reviewer for pointing this out. After trimming the reads from the end and rerunning the whole analysis again, we end up with 5 significant hits (4 TFs and 1 histone mark) in normal and 16 significant hits (12 TFs and 4 histone marks) in tumor. We modified the manuscript so that the numbers on figure 4 are reflected on the manuscript as well.

Point 5: For example, in Methods section 1.3.1, the authors trimmed reads to 49 bp. They should describe how the trimming was performed.

We trimmed reads using the cutadapt software [12] such that reads would be 49 bp in length. For samples with 100 bp read length, we used the following command “*cutadapt 100bpSample.fastq.gz -u -50 -o SampleID_trimmed.fastq.gz -j 6*” and for samples with 75 bp read lengths we used “*cutadapt 100bpSample.fastq.gz -u -26 -o SampleID_trimmed.fastq.gz -j 6*”. We added a paragraph explaining the trimming process in the manuscript section 1.3.1.

Point 6: The authors should explain how to transform TE expression level to "genotype" data.

We thank the reviewer for pointing this out. We correlated TE expression with gene expression using the same premise (linear regression) as for QTL mapping with the QTLtools software. We modified the manuscript and the methods section to avoid any confusion.

Point 7: In Methods section 1.6.4, the authors should explicitly describe the ChIP-seq data sets used.

We thank the reviewer for pointing this out. We added a description for both datasets used in methods section 1.6.4

- **Ensembl Regulatory Build ChIP-seq dataset**

The Ensembl Regulatory Build dataset contains ChIP-seq data from 88 human cell types for a total of 209 transcription factors and 29 histone marks (reference genome GRCh37). The data was downloaded from the ensembl FTP site (http://ftp.ensembl.org/pub/grch37/current/regulation/homo_sapiens/) For each of the TFs and histone marks, we took the union of all peaks together from all 88 cell types. Overlapping peaks were merged together using BEDtools software and “merge” option [13]. This approach allowed us to create an extensive annotation of peaks for 209 TFs and 29 histones genome-wide.

- **Colorectal cancer LoVo cell line ChIP-seq dataset**

We used publicly available ChIP-seq data from colorectal cancer LoVo cell line with accession code [GSE49402](https://www.ncbi.nlm.nih.gov/geo/query/acc.cgi?acc=GSE49402). The dataset comprises 202 TFs and 2 histone marks (reference genome GRCh37). We used BED files containing the coordinates of the peaks for each TF and histone mark for functional enrichment of eQTLs identified in our study.

For gene- and TE- eQTLs in normal and tumor, we used the peak annotation generated from the Ensembl Regulatory Build data to get an extensive comprehension of which TFs regulate the expression of TEs. Regarding the tumor-specific vs shared TE- and gene eQTLs, we used available ChIP-seq data from the colorectal cancer LoVo cell line [14]. We used a cancer specific dataset as we were interested in discovering cancer specific effects.

Point 8: The authors should explicitly list the source of their genotype and RNA-seq data and how to access these data.

We thank the reviewer for pointing this out. The genotype and RNA-seq SYSCOL dataset can be accessed through the European Genome-phenome Archive (EGA, <https://www.ebi.ac.uk/ega/>), accession number EGAS00001000854. We added the source of the genotype and RNA-seq data and how to access them in the revised manuscript.

Point 9: Figure 1A. please check the axis label.

On figure 1A, the panel on the left represents the proportion of expressed TEs per subfamily. For this, we calculated the percentage of TEs that are expressed per subfamily of TEs. The proportion values are very small as only a tiny fraction of TEs are expressed per subfamily. The panel on the right represent the proportion of expressed TE subfamilies compared to all

expressed TEs. For each subfamily of expressed TEs, we calculated the percentage of expressed TEs compared to the other expressed subfamilies of TEs. To make this figure clearer, we would benefit from further instructions from the reviewer as we did not find any error on the axis labels.

Point 10: Figure 6. The legend does not match the figure. Where is figure 6D?

We thank the reviewer for pointing this out. Indeed, there are only three main panels in figure 6. We have corrected the figure legend in the revised manuscript.

Point 11: The authors should host the analysis code for reproduction in a publicly accessible repository such as Github and the TE-eQTLs in a public source that can be downloaded.

We thank the reviewer for pointing this out. All scripts are now available on Github (https://github.com/NLykoskoufis/te_project). Additionally, all results presented in this manuscript can be accessed through the already provided supplementary tables. Below is a table with additional information.

Supplementary Tables	Description
Supplementary Table 1	Enrichment analysis of expressed TEs compared to non-expressed TEs for regulatory regions
Supplementary Table 2	Discovered eQTLs at 5% FDR in normal
Supplementary Table 3	Discovered eQTLs at 5% FDR in tumor
Supplementary Table 4	Replication of SYSCOL normal eQTLs in GTEx colon transverse
Supplementary Table 5	Replication of SYSCOL tumor eQTLs in TCGA-COAD
Supplementary Table 6	Shared eQTL discovery at 5% FDR
Supplementary Table 7	Tumor-specific eQTL discovery at 5% FDR
Supplementary Table 8	Functional enrichment of TE- and gene-eQTLs at 5% FDR in normal
Supplementary Table 9	Functional enrichment of TE- and gene-eQTLs at 5% FDR in tumor
Supplementary Table 10	Functional enrichment of tumor-specific and shared TE-eQTLs
Supplementary Table 11	Functional enrichment of tumor-specific and shared gene-eQTLs
Supplementary Table 12	Causal relationship of eQTL-TE-gene triplets in normal
Supplementary Table 13	Causal relationship of eQTL-TE-gene triplets in tumor
Supplementary Table 14	Replication of normal eQTL-TE-gene triplets in GTEx colon transverse
Supplementary Table 15	Replication of tumor eQTL-TE-gene triplets in TCGA-COAD
Supplementary Table 16	Causal relationship for shared eQTL-TE-gene triplets between normal and tumor
Supplementary Table 17	Causal relationship for the union of eQTL-TE-gene triplets between normal and tumor

Reviewer #2: *This work analyzes the role of transposable elements in gene expression regulation, taking into account genetic variation and DNA methylation. The analyses show interesting global associations, with long lists of genes that might be regulated by TEs. The role in cancer, however, is not well established, since these are observational associations and the Bayesian networks used to infer causality just rely on the strength of the statistical associations.*

Point 1: *Only one dataset has been used, with no attempt to replicate at least the results related to tumor tissue in the larger TCGA dataset. The normal tissue used in the study was obtained from cancer patients, and its gene expression might be altered by the presence of tumor by diverse mechanisms. Again, no attempt has been made to validate the results in other datasets of normal colon tissue. Many of the methods used involve in-house curated databases, which are not made available and difficult the possibility to replicate the results.*

We thank the reviewer for addressing this point and we agree that replicating our normal and tumor findings would increase the robustness of our original discoveries. To this end, we carried out replication analyses and have added the results as well as the approach used in the revised manuscript.

GTEX dataset

We downloaded available data for colon transverse (N=174) and germline genotypes from dbGAP (accession code: phs000424.v8.p2).

Germline genotypes

We used the already filtered VCF file provided by GTEX. The following filters were applied and kept all variants with a MAF $\geq 5\%$, yielding a total of 6,494,417 variants.

RNA-seq dataset

The RNA-seq dataset was treated similarly to SYSCOL RNA-seq data. We first trimmed the reads down to 49bp using cutadapt [12]. Then we mapped and quantified the samples using the exact same approach as for SYSCOL (methods section 1.3.2). Finally, we combined all samples together into a multi-sample bed file and kept all features (TEs, genes) that had less than 50% of missing expression data across all samples, yielding a total of 167,429 TEs and 18,472 genes. Then, we corrected our expression data using the first 3 principal components (PCs) obtained from genotypes, the sex of the samples, the platform they were sequenced and the first 20 PCs obtained from the expression data, for a total of 25 covariates used.

TCGA dataset

We downloaded available germline genotypes and RNA-seq data for colon adenocarcinoma (N=251) from The Cancer Genome Atlas (TCGA) database, accession code phs000178.v11.p8.

Germline genotypes

Germline genotypes were downloaded from the legacy archive GDC portal. We downloaded all germline genotypes for TCGA-COAD in birdseed format. We used birdseed2vcf python tool (<https://github.com/ding-lab/birdseed2vcf>) to convert birdseed to VCF format. We then combined all samples together creating a multi-sample VCF file that we spitted per chromosome and uploaded to the Michigan Imputation Server [15] for imputation and phasing using the Haplotype Reference Consortium (HRC) as reference panel, Eagle v2.4 software [16] for phasing and European (EUR) population. Finally, we merged all chromosome VCFs into a single VCF file and kept variants with a MAF $\geq 5\%$, HWE $> 1e-06$ and $R^2 > 0.3$, yielding a total of 5,511,779 variants.

RNA-seq data

As the read length of TCGA-COAD samples is the same as SYSCOL, we did not need to trim the reads. We mapped, quantified, and filtered our RNA-seq data in a similar way as for SYSCOL and GTEx colon transverse samples yielding a total of 19,376 genes and 75,815 TEs. Expression data was corrected using the same approach as for SYSCOL (methods section 1.3.3) using the first 3 principal components (PC) obtained from genotypes and the first PC obtained from expression data for a total of 4 covariates used.

Replication of eQTL findings

For the replication of our normal and tumor eQTL discoveries, we used the “rep” mode in the QTLtools software [17]. We then used the π_1 metric to estimate the proportion of significance of our eQTLs in GTEx colon transverse. The π_1 is equal to $1 - \pi_0$ where π_0 is the proportion of true null p -values obtained using π_0 est from the Qvalue R package [18].

Replication of the eQTL – TE – gene triplets

We used the same eQTL – TE – gene triplets discovered in normal and tumor and replicated them in GTEx or TCGA-COAD, respectively. We used the exact same approach as previously (methods section 1.8). We then calculate the mean probability of the causal, reactive and independent model. Finally, we compared the percentage of triplets with the same model predicted in both SYSCOL and the replication dataset.

Results

Replication of eQTL discoveries

The figure below represents the p -value distribution of the results. Not all variant-gene or variant-TE pairs were present in the GTEx colon transverse dataset after all filtering steps. Out of the **10,231** TE-eQTLs and **6,955** gene-eQTLs discovered in normal, **8,380 (82%)** and **5,930 (85%)** TE- and gene-eQTLs, respectively were present in the dataset and could be replicated. From the **5,199** TE- and **1,552** gene-eQTLs discovered in SYSCOL tumor, only **3,221 (62%)** TE- and **1,164 (75%)** gene-eQTLs were present in the TCGA-COAD dataset. We observe a high

replication of our original (**Supplementary Figure 7A-C; Supplementary Table 4**) in normal (pi1 TE-eQTLs = **83.1%** pi1 gene-eQTLs = **68.6%**) and tumor (pi1 TE-eQTLs = **88.4%**; pi1 gene-eQTLs = **78.3%**) (**Supplementary Figure 7D-F; Supplementary Table 5**) corroborating our findings.

Point 1 (Supplementary figure 7): Figure representing the p-value distribution of the replication of SYSCOL eQTLs. Panels A to C represent the replications of SYSCOL normal eQTLs in GTEx colon transverse, where (A) represents the p-value distribution of gene- and TE-eQTLs taken together, (B) the p-value distribution of SYSCOL normal TE-eQTLs and (C) the p-value distribution of SYSCOL normal gene-eQTLs. Panels D to F represent the replication of the SYSCOL tumor eQTLs in colon adenocarcinoma (TCGA-COAD), where (D) is the p-value distribution of all gene- and TE-eQTLs taken together, (E) the p-value distribution of SYSCOL tumor TE-eQTLs and (F) the p-value distribution of SYSCOL tumor gene-eQTLs. We observe that in both normal and tumor TE-eQTLs (pi1 normal = 83.1%; pi1 tumor = 88.4%) and gene-eQTLs (pi1 normal = 68.6%; pi1 tumor = 78.3%) have a very high replication in GTEx colon transverse and TCGA-COAD, respectively.

Replication of the eQTL – TE – gene triplet discoveries

We then proceeded with replicating the causal inference of the eQTL – TE – gene triplets to corroborate our findings. We tested the triplets where all three molecular phenotypes were present in either GTEx colon transverse for the SYSCOL normal colon triplets or in TCGA-COAD for the SYSCOL tumor triplets, yielding **9,577 (80%)** triplets and **5,893 (62%)** triplets in common, respectively. We performed BNs similarly to the original discoveries. We observe a

high replication of both normal (**62%** similarity) and tumor (**74%** similarity) results (**Supplementary figure 21; Supplementary Table 14,15**). We believe that the reason the replication of our normal colon eQTLs is lower than for the tumor eQTLs is because of sample size differences between SYSCOL normal colon (N=275) and GTEx colon transverse (N=174) decreasing our statistical power. These results corroborate our findings and highlight that our discoveries are valid.

Point 1 (Supplementary figure 21): This figure represents the replication of the causal inference findings in external datasets. Model probabilities of the causal, reactive and independent models are represented (A) for the 9,577 SYSCOL normal triplets tested in GTEx colon transverse, (B) for the 12,379 normal triplets tested in SYSCOL, (C) for the 9,714 tumor triplets tested in SYSCOL, (D) for the 5,893 SYSCOL tumor triplets tested in TCGA-COAD. (E) represents the percentage of SYSCOL normal triplets with the same model predicted in SYSCOL and in GTEx colon transverse. (F) represents the frequency of substitutions between the normal triplets in SYSCOL and GTEx transverse. (G) represents the percentage of tumor triplets with the same model predicted in SYSCOL and in GTEx colon transverse. (H) represents the frequency of substitutions between the tumor triplets in SYSCOL and TCGA-COAD. We observe a high replication of our causal inference findings in both GTEx colon transverse and TCGA-COAD. Moreover, because the sample size of GTEx colon transverse is smaller (N=174) compared to the one of SYSCOL normal (N=275), we expected a smaller replication of our finding. We observe that in TCGA-COAD, the percentage of replication is higher as the TCGA-COAD and SYSCOL tumor dataset have almost the same sample size ($N_{SYSCOL\ tumor}=276$; $N_{TCGA-COAD}=251$), thus higher statistical power.

Point 2: *The statistical significance of the results are only mentioned in the methods section and, though the methods seem reasonable, reading the results often brings the question "has this been adjusted for multiple comparisons, and how?" As example, only 5 TFs in normal and 15 in tumor were enriched for TE. How many TFs were analyzed? What were the p-values? Authors refer to supplementary tables 6 & 7, but the reported genes in the text are within a long list of significant and non-significant results, and transcription factors are not marked. Supplementary tables don't have a legend to clearly explain their contents.*

We agree with the reviewer that this part needs clarification in the main text and have provided a better explanation in the revised manuscript. We additionally clarified legend text in the supplementary tables and restructured the supplementary tables such that results are outlined in a clearer manner.

Point 3: *The functional enrichment is based on matching ChIP-seq peaks of Ensembl Regulatory Build. All cells are combined for the analysis of eQTLs and TEs, but only cancer LoVo cell line for the comparison of tumor vs shared. Authors could discuss how the diverse cell types combined might affect the functional enrichment results, and the possibility of analyzing ChIP-seq experiments of normal colon cells.*

We agree with the reviewer that using ChIP-seq from normal colon cells would be optimal, but unfortunately we did not encounter a ChIP-seq dataset with such an extensive abundance of TF and histone mark data from normal colon cells.

For the functional enrichment of normal and tumor gene- and TE-eQTLs we indeed took the union of peaks from all available cell types from Ensembl Regulatory build. This allowed us to create a comprehensive annotation of all the ChIP-seq peaks of all the available TFs and histone marks in the dataset. We decided to use this approach to get an extensive overview of which TFs are potentially regulating the expression of TEs and genes in both the normal and tumor state.

In the comparison between tumor-specific versus shared, we looked for TFs that regulate TEs specifically in tumor and not in normal. Thus, we needed a dataset originating from cancer like the LoVo cell line. Using the Ensembl Regulatory Build dataset in that context would not have provided the suitable dataset as cancer-specific ChIP-seq peaks would have not been captured.

Taking the union of all peaks from all available cell types should not impact our analysis as transcription factor binding sites do not change in the genome, they are either active or inactive. In most cases, eQTLs reflect loci where TFs bind [19], thus if an eQTL is active, there is a high chance that TFs are binding at that particular eQTL. Thus, if a ChIP-seq peak overlaps with a specific eQTL, there is a high chance that at this location, a particular TF is binding and regulating the expression of the gene or TE associated with that eQTL.

Point 4: *The discussion sentences are difficult to match to specific results. For example, there is emphasis of findings more relevant in tumors than in normal, but there are 10111 TE-eQTL in normal and 5152 in tumor, a similar ratio for gene-eQTLs. Only enrichment of transcription factors are more often in tumor than normal, but only 15 and 5 were found, respectively.*

We thank the reviewer for this point. We have made extensive modifications to the discussion to better explain our results. It is known that TEs are much more active in tumor than in normal, primarily due to global hypomethylation in cancer driving their expression [20]. However, in our analysis we observed a higher number of eQTLs in normal than in tumor which may sound contradictory. This has to do with the nature of the tumor tissue being much more heterogeneous, thereby increasing the variance in expression data and thus affecting statistical power, leading to fewer eQTLs to discover. Thus, this is solely a statistical problem which a higher sample size could aid in minimizing. To assess the function of these eQTLs, we used functional enrichment analysis. Even though we discovered more eQTLs in normal, we observed that tumor TE-eQTLs were significantly enriched for transcription factors to a greater extent compared to normal TE-eQTLs, thus suggesting that TEs are more active in tumor than in normal.

Reviewer #3: *The authors provide a bioinformatic analysis, which leads to the hypothesis that tumor-specific TE-eQTLs modify the expression of TEs and subsequently alter the expression of an important number of genes in colorectal cancer, including cancer driver genes. The authors also build evidences for the relation of eQTLs, TEs and genes in the regulatory switches occurring from healthy tissue to cancer. The paper is well written, and the emergent hypothesis will be of great interest to the Transposable Elements community. Below, I have noted some comments for the authors to consider when revising their paper for publication in Nature Communications. Page 4 line76-77. As the authors know, filtering for 'uniquely mapped reads' very challenging for young TE families (as explained by Lanciano and Cristofari. Nat Reviews 2020 cited in the bibliography).*

Point 1: *When you find expression of a TE in a donor and not in another due to uniquely mapped reads, how do you know that the donor with no expression simply doesn't have a different SNP variant that you are using to track the uniquely mapped read?*

We thank the reviewer for this point. In 2014, we published a paper that focused on this issue and showed that mapping bias is not an issue regarding exonic regions of the genome [21]. However, the study did not focus on intergenic and repetitive regions where TEs lie. To ascertain whether samples with TEs containing the alternative allele can induce mapping bias, we extracted the reference sequence of all TEs containing a SNP. Then we simulated the alternative allele at each SNP genomic location. We then proceeded in creating two fastq files containing random reads generated using a sliding window approach from the reference sequence of the chosen TEs and the simulated sequence containing the alternative alleles for a total number of reads per TE of 1000. Next, we mapped our simulated reads on the reference genome using the same approach as previously described (methods section 1.3.1-1.3.2). Out of the 50,921 TEs analyzed, 5,771 TEs from our original contained a SNP. We can see that the correlation between read counts obtained when simulating the reference or the alternative allele are highly significantly correlated ($\rho = 0.98$; $p\text{-value} < 2.2e\text{-}16$; **Figure 1**) indicating that the majority of TEs we analyzed are not impacted by mapping bias. Furthermore, as observed in Panousis et al., 2014, this mapping bias does not impact eQTL discovery [21], thus providing assurance that eQTL discovery in our analyses should not be impacted by mapping bias.

Point 1 – Figure 1: Each dot represents a TE containing a SNP from our original 50,921 TEs analyzed. We observe that there is a high and significant correlation between read counts obtained when the TE contains the reference allele or the alternative allele.

Point 2: And when you see more expression in a tumor sample than in a healthy matched tissue, how do you know that this is not due to the TE increasing its copy number in tumor samples (active ones through transposition, and older ones through rearrangements/duplications)?

We thank the reviewer for this excellent point. It is known in colorectal cancer that patients with microsatellite instability (MSI) tend to have more rearrangements/duplications of the genome compared to patients with microsatellite stable (MSS) [22]. Thus, we compared the levels of expression of TEs between MSS (N=224) and MSI samples (N=52). We then performed multiple test correction using the Qvalue R package with an FDR threshold of 5%. Only 3 TEs of 50,921 tested were found to have a differential expression level between MSI and MSS samples (**Figure 1-2**). These results suggest that rearrangements or duplications of the genome should not impact the differences in TE and gene expression we are observing between normal and tumor states.

Point 2 – Figure 1: P-value distribution for the 50,921 TEs tested for difference in expression levels between MSI and MSS samples.

Point 2 – Figure 2: Boxplots of the 3 TEs that show significant differences in expression levels at 5% FDR between MSI and MSS samples.

Transposition events can indeed increase the copy number of TEs, thus increase the expression levels in tumor compared to normal. Unfortunately, with the current dataset it is impossible to assess transposition events as Whole Genome Sequencing (WGS) data is required for accurate transposition event discovery. More comprehensive assessment of the extent of the effect of transposition on TEs is needed in larger sample sets with WGS data.

Point 3: I consider that the authors should validate their results with unique reads mapping the junctions TE-genome for the younger TEs as the bona fide way to uniquely map the expression of a specific TE locus (particularly in the case of young L1s).

We thank the reviewer for this point. Indeed, younger TEs are more difficult to uniquely map as their sequence will be similar between the different integrants due to shorter evolutionary time for accumulating sufficient mutations to distinguish between the different integrants.

We carried out the quantification approach Reviewer #3 suggested for all TEs (young and older ones). Specifically, for each TE, we counted only reads that overlapped between the TE and the flanking genomic region and kept reads that had an overlap in the flanking genomic sequence or TE sequence of at least 8 or 16 bases (a combination of a sequence of 8 or 16 nucleotides should not be overrepresented in the genome). We then applied the same filtering steps as previously mentioned (methods section 1.3.2). In brief, we combined all single sample quantification into a multi-sample matrix in which we only kept TEs where the sum of reads was higher than the total number of samples and where there was at least one sample with more than 20 reads. Next, we filtered out any TEs with more than 50% of missing expression (read count of 0). Even though the expression levels between the TE-genome junction quantification approach and original approach (using featureCounts) are significantly correlated for all samples (**Figure 1-2**), the number of TEs that surpassed all filtering steps is very small (**Figure 3**): only a fraction of TEs with more than 20 overlapping reads. Focusing on the younger TEs (i.e., L1s), we again observe that the number of TEs with at least 20 reads overlapping them is very small (**Figure 4**). In conclusion, this approach is too stringent to be used with the current dataset. Additionally, quantifying young and older TEs differently will create a bias as one method is more stringent than the other.

Point 3 – Figure 1: Spearman correlation between read counts obtained from the TE-genome junction quantification approach and the original one using featureCounts. (A) represents the spearman coefficient (ρ) for the spearman test between read counts obtained with the TE-genome junction approach with a minimum of 16 bases overlap on the genome and TE and the original one using featureCounts **(B)** represents the spearman coefficient (ρ) for the spearman test between read counts obtained with the TE-genome junction approach with a minimum of 8 bases overlap on the genome and TE and the original one using featureCounts. We observe that all 551 samples are significantly correlated between the two quantification approaches.

Point 3 – Figure 2: Spearman correlation between original quantification using featureCounts or TE-genome junction approach. We observe that counts are significantly correlated between both quantification approaches (featureCounts vs TE-genome junction).

Point 3 – Figure 3: TEs passing read threshold for the different quantification approaches used. We observe that with the TE-genome junction approach the amount of TEs with at least one read overlapping them decreases substantially compared to featureCounts making it too stringent with this type of dataset.

Point 3 – Figure 4: Young TEs (L1s and LTR) passing read threshold for the different quantification approaches used. This figures represent the number of young TEs, L1s in a normal sample in (A) and (B) in a tumor sample and LTRs in (C) a normal sample and (D) in a tumor sample. We observe the same as in point 3 – figure 3.

Point 4: Fig 1.A. It would be really informative to analyse the proportion of TE subfamilies that are expressed in tumor and normal tissue in different graphs to estimate the differences in expression in both types of tissues.

We thank the reviewer for this suggestion. We processed normal and tumor samples separately and compared the proportion of TE subfamilies that are expressed in tumor and normal tissues. We then tested whether there is any significant difference between TE subfamilies using a Fisher's exact test and corrected for multiple testing. At 5% FDR, there is a significantly higher proportion of expressed MIR from the SINE family in tumor compared to normal. On the other hand, there is a significantly higher proportion of expressed ERVK from the LTR family in normal compared to tumor. For all the other subfamilies with expressed TEs in our dataset, we did not observe any significant difference between the proportion of expressed TEs in normal and tumor.

Point 4 : Figure representing the proportion of expressed TE subfamilies stratified by tissue (i.e., normal or tumor).

Point 5: Page 5 line 102. The authors claim that the smaller number of eQTL per TE compared to genes could be due to smaller evolutionary time of TE regulatory landscapes. The authors could test this hypothesis calculating the number of eQTL for TEs groups that contain a succession of subfamilies with a wide and well established age scale (LINEs for example).

We thank the reviewer for this suggestion. We used TE age scales from the DFam database [23] and calculated the number of independent eQTLs per TE subfamily. Then, as various TE

subfamilies had the same age scale, we stratified the data per age scale and estimated the mean number of independent eQTLs. The figure below represents the mean number of eQTLs per age scale. We did not find any significant correlation ($\rho = -0.34$; p -value = 0.22) indicating that older TEs do not have on average more independent eQTLs than younger ones.

We do want to point out to that in the original manuscript we compared genes versus TEs and not between specific TE subfamilies. The evolutionary time of the various TE subfamilies in the human genome is just a mere subset of the evolutionary time between genes and TEs. The various TE subfamilies have not been in our genome for a sufficient amount of time (compared to genes) which is why we do not observe a significant correlation between age of TEs in the genome and number of independent eQTLs.

Point 5: Figure representing the mean number of independent eQTLs as a function of the estimated age of transposable elements. We observe that there is no significant correlation (p -value = 0.22) between the number of independent eQTLs per TE and the age of TEs.

Point 6: Page 7 line 152. The identification TFs involved in the expression of tumor-specific eQTLs has been done with CHIP-seq data from LoVo colorectal cancer and the authors suggest that changes in methylation patterns allow more accessibility of those TFs in tumor. To increase the robustness of the proposed mechanism, a CHIP-qPCR in colorectal cancer and primary colon cell lines would be highly recommendable to validate these results. Another, no-wet lab option to demonstrate that the TE-eQTLs are more accessible to TFs will be the comparison with ATAC-seq data from colorectal cancer/normal tissues.

We downloaded available ATAC-seq data from Encode for normal Colon sigmoid and cancer ATAC-seq data from TCGA-COAD [24]. ATAC peaks were lifted over from hg38 to hg19 using the liftOver software from UCSC. We only kept ATAC peaks that were exclusive to the cancer data and absent from the normal ATAC-seq dataset as these are more probable to be tumor-specific. We then created a 2x2 contingency table and used Fisher’s exact test to assess whether tumor-specific TE-eQTLs are enriched for tumor-specific ATAC peaks compared to shared TE-eQTLs.

	Tumor-specific TE-eQTLs	Shared TE-eQTLs
Overlapping tumor specific ATAC peaks	2	3
Not overlapping tumor specific ATAC peaks	427	522

The fisher exact test being not significant (p-value=1, Odds-ratio = 0.815) indicates that tumor-specific TE-eQTLs are not enriched for the tumor-specific ATAC-seq peaks. This can be due to the small number of tumor-specific TE-eQTLs and shared eQTLs discovered, thus decreasing power. Ideally, for this type of analysis, we would need to perform ATAC-seq experiment on the SYSCOL samples and then perform the analysis suggested by Reviewer #3.

Point 7: Page8 line185. The authors detect that the “reactive model” is more frequently for intronic TEs. The authors have not elaborated if this could be a consequence of the TE being transcribed within the gene rather than as a consequence of the TE promoter stimulation.

The reactive model is when the eQTL variant impacts gene expression which then results in TE expression. In both normal and tumor, we found that the most probable model was the reactive model where the triplet is constituted within an intronic TE than an intergenic TE. As suggested, we do believe that the expression levels of these intronic TEs are a consequence of the transcription of the gene and not the consequence of the transcription from the TE promoter. We thank the reviewer for this point and have updated the revised manuscript accordingly.

Point 8: It is expected a 50% distribution of sense/antisense TE insertion in Introns (and for LINE-1s, this appears to be even bias towards antisense), so it could be interesting to select antisense intronic TEs and detect if the TE reads in these cases are in sense respect to the gene or in sense respect to the TE (if RNA-seq data has been prepared with stranded-kits to detect transcriptional direction).

This is an excellent point addressed from the reviewer but as SYSCOL RNA-sequencing data is not stranded, we cannot look further into this.

Point 9: Page 10 line 238. It appears that the tumor specific TE-eQTLs contribute to tumorigenesis by impacting genes through TEs. In fact, the authors shown that the expression of 34 CDGs was positively correlated with the expression of TEs exclusively in tumor. Ideally, experimental validation would be necessary for this kind of associations. Others authors have already CRISPR deleted TEs (Chuong, Elde and Feschotte, Science 2016). Authors could remove the TE in cell lines and verify the succession of events that they are proposing here.

We thank the reviewer for this suggestion. Testing whether tumor-specific TE-eQTLs contribute to tumorigenesis by impacting CDGs through TEs remains out of scope of the current manuscript at this point.

REFERENCES

1. Liao, Y., G.K. Smyth, and W. Shi, *featureCounts: an efficient general purpose program for assigning sequence reads to genomic features*. *Bioinformatics*, 2014. **30**(7): p. 923-930.
2. Sexton, C.E. and M.V. Han, *Paired-end mappability of transposable elements in the human genome*. *Mob DNA*, 2019. **10**: p. 29.
3. Kong, Y., et al., *Transposable element expression in tumors is associated with immune infiltration and increased antigenicity*. *Nature communications*, 2019. **10**(1): p. 1-14.
4. Jin, Y., et al., *TEtranscripts: a package for including transposable elements in differential expression analysis of RNA-seq datasets*. *Bioinformatics*, 2015. **31**(22): p. 3593-3599.
5. Ainsworth, H.F., S.Y. Shin, and H.J. Cordell, *A comparison of methods for inferring causal relationships between genotype and phenotype using additional biological measurements*. *Genetic epidemiology*, 2017. **41**(7): p. 577-586.
6. Scutari, M., *Learning Bayesian Networks with the bnlearn R Package*. 2010, 2010. **35**(3): p. 22.
7. Bowden, R. and D. Turkington, *Instrumental Variables*. Cambridge University Press. Cambridge, UK, 1984.
8. Didelez, V. and N. Sheehan, *Mendelian randomization as an instrumental variable approach to causal inference*. *Statistical methods in medical research*, 2007. **16**(4): p. 309-330.
9. Viñuela, A., et al., *Genetic analysis of blood molecular phenotypes reveals regulatory networks affecting complex traits: a DIRECT study*. medRxiv, 2021.
10. Delaneau, O., et al., *Chromatin three-dimensional interactions mediate genetic effects on gene expression*. *Science*, 2019. **364**(6439).
11. Gutierrez-Arcelus, M., et al., *Passive and active DNA methylation and the interplay with genetic variation in gene regulation*. *Elife*, 2013. **2**: p. e00523.
12. Martin, M., *Cutadapt removes adapter sequences from high-throughput sequencing reads*. *EMBnet. journal*, 2011. **17**(1): p. 10-12.
13. Quinlan, A.R. and I.M. Hall, *BEDTools: a flexible suite of utilities for comparing genomic features*. *Bioinformatics*, 2010. **26**(6): p. 841-2.
14. Yan, J., et al., *Transcription factor binding in human cells occurs in dense clusters formed around cohesin anchor sites*. *Cell*, 2013. **154**(4): p. 801-13.
15. Das, S., et al., *Next-generation genotype imputation service and methods*. *Nature genetics*, 2016. **48**(10): p. 1284-1287.
16. Loh, P.-R., et al., *Reference-based phasing using the Haplotype Reference Consortium panel*. *Nature genetics*, 2016. **48**(11): p. 1443-1448.
17. Loh, P.-R., P.F. Palamara, and A.L. Price, *Fast and accurate long-range phasing in a UK Biobank cohort*. *Nature genetics*, 2016. **48**(7): p. 811-816.
18. Dabney, A., J.D. Storey, and G. Warnes, *qvalue: Q-value estimation for false discovery rate control*. R package version, 2010. **1**(0).
19. Flynn, E.D., et al., *Transcription factor regulation of eQTL activity across individuals and tissues*. *PLoS genetics*, 2022. **18**(1): p. e1009719.
20. Jang, H.S., et al., *Transposable elements drive widespread expression of oncogenes in human cancers*. *Nature genetics*, 2019. **51**(4): p. 611-617.

21. Panousis, N.I., et al., *Allelic mapping bias in RNA-sequencing is not a major confounder in eQTL studies*. *Genome Biol*, 2014. **15**(9): p. 467.
22. Boland, C.R. and A. Goel, *Microsatellite instability in colorectal cancer*. *Gastroenterology*, 2010. **138**(6): p. 2073-2087. e3.
23. Storer, J., et al., *The Dfam community resource of transposable element families, sequence models, and genome annotations*. *Mobile DNA*, 2021. **12**(1): p. 1-14.
24. Corces, M.R., et al., *The chromatin accessibility landscape of primary human cancers*. *Science*, 2018. **362**(6413): p. eaav1898.

REVIEWER COMMENTS

Reviewer #1 (Remarks to the Author):

The authors have addressed several concerns, but not all of them.

1. It is quite confusing with the new supplement figure 5 that most QTLs are located far from their TSS, which seems opposite to their Figure 2A-B conclusion.

2. The authors should also provide their analysis data, otherwise it is hard to evaluate their analysis without providing their inputs. This is important especially considering the authors have already found many errors in their original analysis such as TF result.

Reviewer #2 (Remarks to the Author):

In this revised version, authors have appropriately responded to the critiques.

DNA methylation data probably should point to EGAD00010001888

Reviewer #3 (Remarks to the Author):

I appreciate the efforts from the authors to address most of the concerns initially raised by me. However, the limitations of the study that specially affect young and active TE subfamilies, such as L1Hs that are highly expressed and mobilized in colorectal cancer, are still unsolved (I did not mean the entire L1 family). The huge undestimation of the expression of those subfamilies coming from the use of “uniquely mapped reads” and the fact that reads from new tumor-specific insertions were not considered in the analysis should be further discussed in the discussion. The authors should clarify that the accuracy of this analysis is satisfactory exclusively for the expression of old TE families.

Manuscript NCOMMS-21-48187A “Transposable elements mediate genetic effects altering the expression of nearby genes in colorectal cancer”

Point by point responses to reviewers – Second round

Reviewer #1

The authors have addressed several concerns, but not all of them.

1. It is quite confusing with the new supplement figure 5 that most QTLs are located far from their TSS, which seems opposite to their Figure 2A-B conclusion.

Figure 2A-B represent the strength of the eQTL ($-\log_{10}$ p-value) as a function of the distance of the eQTL variant to the Transcription Start Site (TSS) of the TE or gene. The eQTL window is ± 1 Mb from the TSS meaning that eQTL variants downstream of the TSS will have a positive distance and eQTL variant upstream will have a negative distance. Supplementary Figure 5 represents the distance of the eQTL variant from the TSS of the gene or TE \log_{10} scaled and in absolute values, meaning that all negative values become positive. This leads to more eQTLs being located far from the TSS. The purpose of supplementary figure 5 is to show whether there is a difference in the distance of the eQTL variants between TE-eQTLs and gene-eQTLs and we had to take the absolute values to do so. As observed on supplementary figure 5, TE-eQTLs are significantly closer to the TSS of TEs than gene-eQTLs are to the TSS of genes. We made the y label of supplementary figure 5 clearer so that reviewers and readers understand it better. From the figure below, we can observe that as mentioned in the manuscript, eQTLs for TEs and genes are clustering very near the TSS (at position 0).

Figure representing the distance to TSS for TE- and gene-eQTLs in both normal and tumor without taking the absolute values. We can see that as mentioned in the main manuscript, eQTLs tend to cluster close to the TSS.

2. The authors should also provide their analysis data, otherwise it is hard to evaluate their analysis without providing their inputs. This is important especially considering the authors have already found many errors in their original analysis such as TF result.

As the SYSCOL data is protected data, we cannot provide the input files but reviewers can access it through the accession number EGAC00001000204. All results presented in the manuscript can be accessed in the supplementary excel sheets provided with the manuscript.

We would like to point out to the reviewer that there were not “many errors” in the original analysis. The main “error” was incorrectly trimming reads from the 5’ end instead 3’ but as showed results are not affected by this and all final conclusions remain the same. There was no mistake with the functional enrichment of eQTLs. What the reviewer is referring to is the fact that there was a difference in the number of significant hits between the main text and figure 4 regarding functional enrichments. We previously commented that this was due to

the fact that we had decided to report only significant hits of TFs in the main text but show all significant hits in figure 4 (Transcription factors + histone marks). However, we agreed that this was very misleading thus, we decided to report all significant hits (TFs + histone marks) in both figure 4 and the main text to avoid further confusion.

Reviewer #2

In this revised version, authors have appropriately responded to the critiques. DNA methylation data probably should point to EGAD00010001888.

We thank the reviewer for pointing this to us. We added this in the main manuscript under the data availability statement.

Reviewer #3

I appreciate the efforts from the authors to address most of the concerns initially raised by me. However, the limitations of the study that specially affect young and active TE subfamilies, such as L1Hs that are highly expressed and mobilized in colorectal cancer, are still unsolved (I did not mean the entire L1 family). The huge undestimation of the expression of those subfamilies coming from the use of “uniquely mapped reads” and the fact that reads from new tumor-specific insertions were not considered in the analysis should be further discussed in the discussion. The authors should clarify that the accuracy of this analysis is satisfactory exclusively for the expression of old TE families.

We agree with reviewer #3 that our study is biased towards the expression of older TEs. Indeed, younger TE subfamilies like L1HS are highly expressed and mobilized in colorectal cancer. However, in the current dataset we have, it is very difficult to properly quantify such molecular elements as the sequence between the various integrants will be highly similar. We believe that studying younger TE families, like the L1Hs is possible either by looking at the expression levels of all expressed L1Hs by aggregating their expression or by using long read sequencing technologies. We added a paragraph in the discussion of the manuscript highlighting this point.

REVIEWERS' COMMENTS

Reviewer #1 (Remarks to the Author):

The reviewer has well addressed my concerns.

Reviewer #3 (Remarks to the Author):

The authors have addressed most of the main points carefully. I have no further comments at this time.

Manuscript NCOMMS-21-48187A “Transposable elements mediate genetic effects altering the expression of nearby genes in colorectal cancer”

Point by point responses to reviewers – Third round

Reviewer #1 (Remarks to the Author):

The reviewer has well addressed my concerns.

Reviewer #3 (Remarks to the Author):

The authors have addressed most of the main points carefully. I have no further comments at this time.